# A smart adhesive Janus hydrogel for non-invasive cardiac repair and tissue adhesion prevention

Yutong He [1,2], Qian Li[2,3], Pinger Chen [2,3], Qixiang Duan [2,3], Jiamian Zhan[2,3], Xiaohui Cai[4], Leyu Wang[2,5], Honghao Hou [2,3,6] ✉ & Xiaozhong Qiu [1,2,6] ✉

Multifunctional hydrogel with asymmetric and reversible adhesion characteristics is essential to handle the obstructions towards bioapplications of trauma removal and postoperative tissue synechia. Herein, we developed a responsively reversible and asymmetrically adhesive Janus hydrogel that enables on-demand stimuli-triggered detachment for efficient myocardial infarction (MI) repair, and synchronously prevents tissue synechia and inflammatory intrusion after surgery. In contrast with most irreversibly and hard-to-removable adhesives, this Janus hydrogel exhibited a reversible adhesion capability and can be noninvasively detached on-demand just by slight biologics. It is interesting that the adhesion behaves exhibited a molecularly encoded adhesion-adaptive stiffening feature similar to the self-protective stress–strain effect of biological tissues. In vitro and in vivo experiments demonstrated that Janus hydrogel can promote the maturation and functions of cardiomyocytes, and facilitate MI repair by reducing oxidative damage and inflammatory response, reconstructing electrical conduction and blood supply in infarcted area. Furthermore, no secondary injury and tissue synechia were triggered after transplantation of Janus hydrogel. This smart Janus hydrogel reported herein offers a potential strategy for clinically transformable cardiac patch and anti-postoperative tissue synechia barrier.

Myocardial infarction (MI) is a leading cause of death worldwide[1]. In recent years, diverse technologies, including injectable biomaterials and cardiac patches, have been developed to rebuild the function of infarcted myocardium[2,3]. Nevertheless, either intraventricular/catheter injections or patch sutures can cause new damage to the delicate myocardium and lead to tissue synechia between the heart and surrounding tissue[4]. Based on these reasons, it is urgent to explore benign strategies that are more suitable for clinical application.

A multifunctional hydrogel cardiac patch with Janus adhesive characteristics and specific functions on asymmetric sides is most fantastic and promising for simultaneously achieving the properties and dual-functions of the MI repair and preventing postoperative the tissue synechia and secondary trauma. Janus hydrogels with asymmetric characteristics on opposite sides have attracted increasing interest due to their great potentials for various biomedical applications including tissue scaffolds, soft actuators, micro surgery robotics,

[1]The Fifth Affiliated Hospital, Southern Medical University, Guangzhou, Guangdong 510900, People's Republic of China. [2]Guangdong Provincial Key Laboratory of Construction and Detection in Tissue Engineering, Guangzhou, Guangdong 510515, People's Republic of China. [3]School of Basic Medical Science, Southern Medical University, Guangzhou, Guangdong 510515, People's Republic of China. [4]School of Pharmaceutical Science, Southern Medical University, Guangzhou, Guangdong 510515, People's Republic of China. [5]Biomaterials Research Center, School of Biomedical Engineering, Southern Medical University, Guangzhou, Guangdong 510515, People's Republic of China. [6]These authors contributed equally: Honghao Hou, Xiaozhong Qiu. ✉ e-mail: ss.hhh89@hotmail.com; qqiuxzh@163.com

hydrogel patches, and wearable electronics[5–7]. However, most of the reported works related to asymmetric hydrogels focused on the repair of abdominal wall and gastric perforation defect, or preventing post-surgical adhesion formation. Considering the elusive complexity and high challenge of cardiac tissue and repair of injury, it, therefore, remains a great challenge to develop a Janus hydrogel for MI repair and anti-tissue-synechia till now.

It is well known that cardiac tissue is replaced by scar tissue after infarction, the electrical pulse signal in the infarcted area is blocked, and with the progress of ventricular remodeling, the ventricular wall becomes thinner, and the ventricular stress increases. Providing myocardial tissue with mechanical support and reconstructing electrical conduction in MI have been proved an effective approach to promote cardiac function after MI[8–10]. However, most of the adhesive hydrogels reported have poor mechanical properties and electrical conductivity, which are difficult to meet the requirements of cardiac patch[11,12]. In addition, a lot of reactive oxygen species (ROS) is produced after MI, which further promotes inflammation, necrosis, and fibrosis, worsening the pathological process[13–16]. Continuous oxidative damage and inflammatory response lead to apoptosis and injury of cardiomyocytes (CMs) and inhibit normal repair and regeneration of the infarcted heart. By reducing the levels of ROS and inflammatory factors in tissues, cardiac patch can prevent the adverse cell signal cascade after MI, reduce the apoptosis and necrosis of CMs, and inhibit the occurrence of ventricular remodeling. Nevertheless, most adhesive hydrogels mainly focus on adhesive properties, and little consideration is given to the anti-inflammatory and antioxidant properties of the materials, fail to reduce ROS and inflammatory levels in MI tissues.

Although traditional Janus adhesive has the asymmetric function of adhesion and anti-adhesion, which can avoid secondary damage and tissue synechia caused by suturing, these adhesives lack the electrical conductivity, mechanical properties, antioxidant and anti-inflammatory properties required for MI repair. Therefore, the development of a multifunctional cardiac patch with Janus adhesion properties and asymmetric side-specific functions is the most ideal, which is expected to achieve both the characteristics and dual functions of MI repair, and prevent postoperative tissue adhesion and secondary injury. In addition, most adhesive hydrogels are irreversible, and they are difficult and excruciating to remove them once they are attached to tissues, which may easily cause tissue damage. Therefore, it is also very important and urgent for tissue engineering to develop a reversible adhesive hydrogel that can be removed non-invasively and on demand. However, up to now, a Janus adhesive cardiac patch with simultaneous multiplex-functions of MI repair, preventing postoperative tissue synechia and removed on demand, has not been reported yet.

Herein, to achieve perfect therapeutic performance and satisfy the clinical requirements, we report an asymmetrically adhesive Janus hydrogel that quickly adheres to the surface of the heart and enabling on-demand stimuli-triggered detachment, and prevents tissue adhesion after surgery. The atop layer exhibited the anti-cell adhesion and non-fouling properties, therefore well-preventing tissue adhesion. Conversely, the bottom hydrogel layer not only can stably adhere to cardiac tissue and easily be removed on demand, but also play a restorative role for myocardial infarction. In addition, the hydrogel processed appropriate properties and great potential for cardiac patches with good electrical conductivity, mechanical properties, anti-inflammatory, and antioxidant properties, which can offer mechanical support, reconstruct the electrical conduction after MI, reduce oxidative damage and inflammatory response after MI, and promote the recovery of cardiac function and angiogenesis. These unique properties and asymmetrical versatility would be very promising to surmount the difficulties of the practical administration of cardiac patch.

## Results

### Design, characterization, and physicochemical properties of CPAMC/PCA Janus hydrogel potential for cardiac patch

To fulfil the multiple properties and functionalities of promoting the MI repair and pro-healing effect, and averting the secondary trauma on a single cardiac patch, as depicted in Fig. 1, we designed and developed a Janus adhesive double-layer hydrogel (named CPAMC/PCA hydrogel) with asymmetric adhesion and anti-synechia on opposite sides via an interfacial co-polymerization strategy introducing an anti-cell-adhesion hydrogel layer atop a conducting, elastic and adhesive hydrogel. The bottom hydrogel is proposed to ensure a good adhesion ability onto wet cardiac tissue and offer a mechanical-electrical and ROS scavenging environment for MI repair. Firstly, a 3D porous elastic and electrical-conducting hydrogel bottom layer with matched electromechanical property is the principal key for functional cardiac patch with suitable mechanical supporting and electrical conduction integration. A mussel-inspired adhesive ionic hydrogel was prepared as bottom layer (named CPAMC hydrogel) by constructing a redox-responsive PAA/PEI/CNC-CHO/CA interpenetrating network containing covalent cross-linking and noncovalent interactions using acrylic acid (AA), polyethylenimine (PEI), aldehyde cellulose (CNC-CHO) with N, N′-bis(acryloyl) cystamine (BAC) as dynamic cross-linker, which can response to ROS enriched in infarcted myocardial region (Supplementary Fig. 1 and Supplementary Table 1). Furthermore, for tough wet adhesion capability, we introduced an ionization promoter 3-sulfonic acid propyl methyl acrylic acid potassium (MASEP) and caffeic acid (CA) with catechol groups for enhanced interfacial interactions, including hydrogen bonding, electrostatic interaction, and cation−π interaction, between tissue and hydrogel patch[17,18]. These multiplex dynamic noncovalent interactions based on the catechol groups, benzene rings of CA, carboxyl of AA and amino of PEI and interlaced interpenetration and physical entanglement of PEI and PAA polymeric chains enable the CPAMC hydrogel with high adhesion strength, and mechanical and electrochemical stability. In addition, due to the presence of residual quinone groups and aldehyde in CPAMC hydrogel, which can react with nucleophilic groups of tissue surfaces (such as $-NH_2$ and −SH) via Michael-type addition and Schiff base formation, it is expected that the adhesion capacity of CPAMC hydrogel can be further improved resulted from strong adhesion by forming another interpenetrating polymer network between tissue and the adhesive hydrogel substrate. Conversely, the top-layer (named PCA) hydrogel was designed with AA and carboxylated cellulose (CNC-COOH) as components, anti-cell-adhesive polyethylene glycol diacrylate (PEGDA) as crosslinking agent to construct the top layer with the anti-cell adhesion and non-fouling properties, therefore well-preventing tissue synechia. We obtained a double layer hydrogel by in-situ polymerizing and gelatinizing a second layer hydrogel atop the interface of a pre-semicured bottom CPAMC hydrogel solution (Supplementary Fig. 1). The multiple interactions between the two layers including adhesion, electrostatic adsorption, dynamic crosslinking, and permanent crosslinking (free radical polymerization) made the layers connect tightly. Importantly, due to its special structure and composition, the obtained Janus bilayer hydrogels exhibit excellent adhesion and anti-synechia asymmetry at targeted tissue sides (Fig. 1). The design concept of asymmetric double-layer hydrogel will provide a new strategy for the development of a variety of tissue engineering scaffolds.

To meet the adhesion and elasticity requirements of cardiac patch, acrylic acid with a large number of carboxyl groups and polyethylene imine with a large number of amino groups were selected as the skeleton of the adhesive layer hydrogel, and BAC was used as a dynamic crosslinking agent to construct a hydrogel with good adhesion, elasticity and stimulated removal capacity. In order to endow hydrogels with suitable conductivity and reconstruct

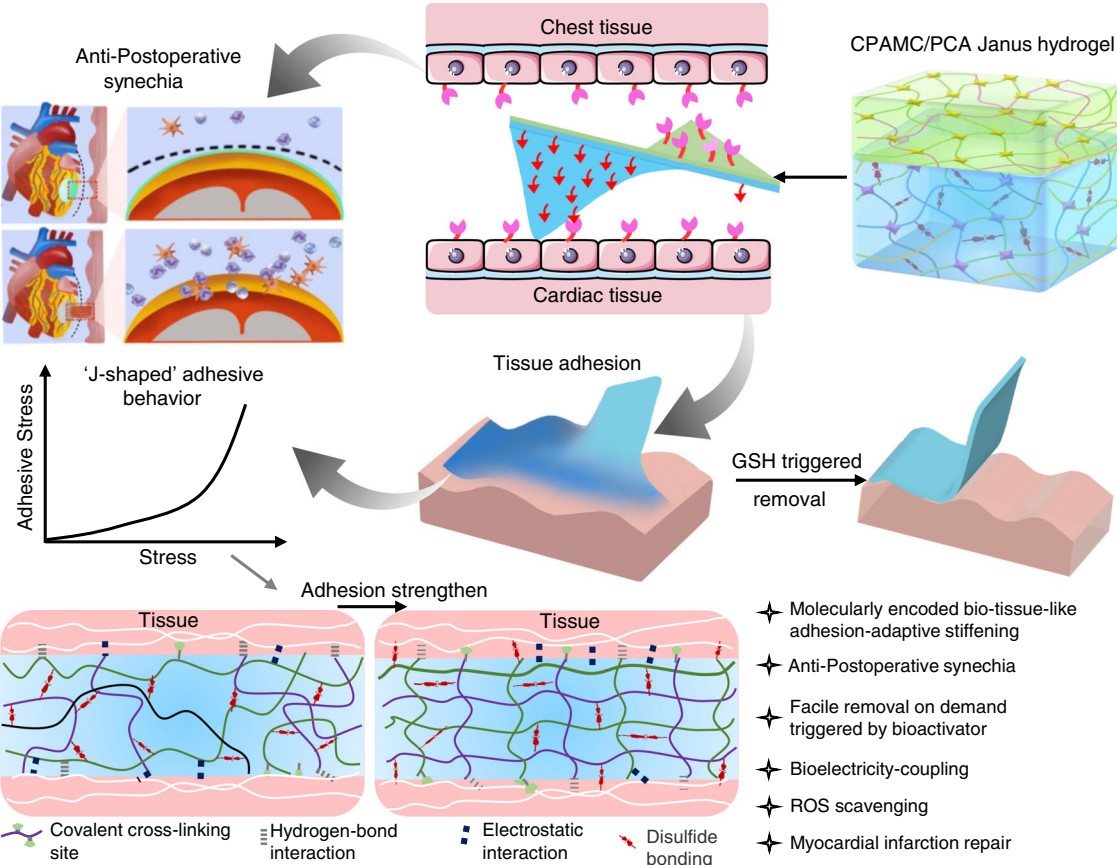

**Fig. 1 | Schematic illustration of the smart Janus adhesive and on-demand removable CPAMC/PCA hydrogel as a multifunctional engineered cardiac tissue patch (ECP) to repair rat's myocardial infarction (MI).** The Janus CPAMC/PCA hydrogel can quickly adhere to the wet heart tissue and enable on-demand stimuli-triggered detachment, and simultaneously prevent post-tissue synechia. CPAMC hydrogels have greatly improved their adhesion properties through interpenetrated network containing multiplex covalent and non-covalent interactions, and interlaced interpenetration and physical entanglement of macromolecular chains. With the increase of adhesion time, the surface water is excluded and solidified with ROS responsive hydrogel system, the adhesion behaves exhibited the arresting feature with molecularly encoded adhesion-adaptive stiffening similar to the self-protective stress–strain effect of biological tissues. CPAMC/PCA Janus adhesive hydrogel can be used to treat MI by reconstructing electrical conduction and removing excess ROS in MI region. CPAMC: C, aldehyde cellulose (CNC-CHO); P, polyethylenimine (PEI); A, acrylic acid; M, 3-sulfonic acid propyl methyl acrylic acid potassium (MASEP); C, caffeic acid (CA). PCA: P, polyethylene glycol diacrylate (PEGDA); C, carboxylated cellulose (CNC-COOH); A, acrylic acid.

---

electrical-coupling microenvironment in infarcted zone, an ionization promoter 3-sulfonic acid propyl methyl acrylic acid potassium (MASEP) were introduced, in which sulfonate groups were proven to be beneficial for angiogenesis and polarization of M1 to M2 type macrophages[19]. A number of studies show that the catechol structure in the protein secreted by mussels can firmly adhere to the reef in the ocean[20–25]. Therefore, caffeic acid with catechol structure was introduced to improve the adhesion of hydrogels to wet surface tissues. The heart is in a dynamic cycle of contraction and extension for a long time, so the cardiac patch needs to have good fatigue resistance to adapt to the function of the heart, so as to avoid the fracture of the patch. With regard to enhancing the fatigue property hydrogels, integration of nanofillers into hydrogel matrix is usually considered as a promising strategy for their prominent contributions to prevent crack diffusion and stress transfer between soft matrix and hard nanofillers effectively. The functional cellulose nanocrystals (CNCs) anchoring with sulfonate and aldehyde groups have excellent performance in tuning mechanical properties and enhanced adhesion of hydrogels due to the for better due to their good biocompatibility, excellent mechanical properties, and variability with physical interlocking, abundant intermolecular interaction sites[26–28].

Compared to CNCs from other sources, CNCs isolated from the tunic of sea squirts exhibit higher aspect ratio, crystallinity, and modulus, enabling the resultant hydrogels with better fatigue resistance and other mechanical properties[29]. Therefore, homogeneous CNCs were obtained from sea squirt by dissolving amorphous cellulose in sulfuric acid and preserving the crystalline region of cellulose. From TEM observation, we see that a CNC with a diameter of several nanometers and a length of about 2 microns was successfully prepared (Supplementary Fig. 2a). In addition, it was found that modification of CNCs with sulfonate and aldehyde groups could endow hydrogels with certain adhesion[30]. Therefore, we further modified CNCs to obtain aldocellulose nanocrystals (CNC-CHO) through being oxidized by NaIO$_4$ to enhance the adhesion of hydrogels with bio-tissues. We confirmed the chemical structure of CNC-CHO using the FT-IR (Supplementary Fig. 2b). Compared with that of pure CNC, the FT-IR spectrum of CNC-CHO showed an obvious absorption peak of aldehyde group at 1600 cm$^{-1}$, implying the successful synthesis of aldocellulose[31].

The adhesive CPAMC nanocomposite hydrogels were fabricated by free radical polymerization as illustrated in Supplementary Fig. 1. We constructed an interpenetrating polymer network consisting of a P(AA-co-MASEP) covalently cross-linked network and an ionically or hydrogen-bond cross-linked network (PEI/caffeic acid/CNC-CHO) with ammonium persulfate (APS) and BAC as initiator and chemical cross-linker, respectively. FT-IR analysis was applied to verify the chemical structures of three different hydrogels. As shown in Supplementary Fig. 3, all hydrogels showed the absorption peaks of amide bonds at

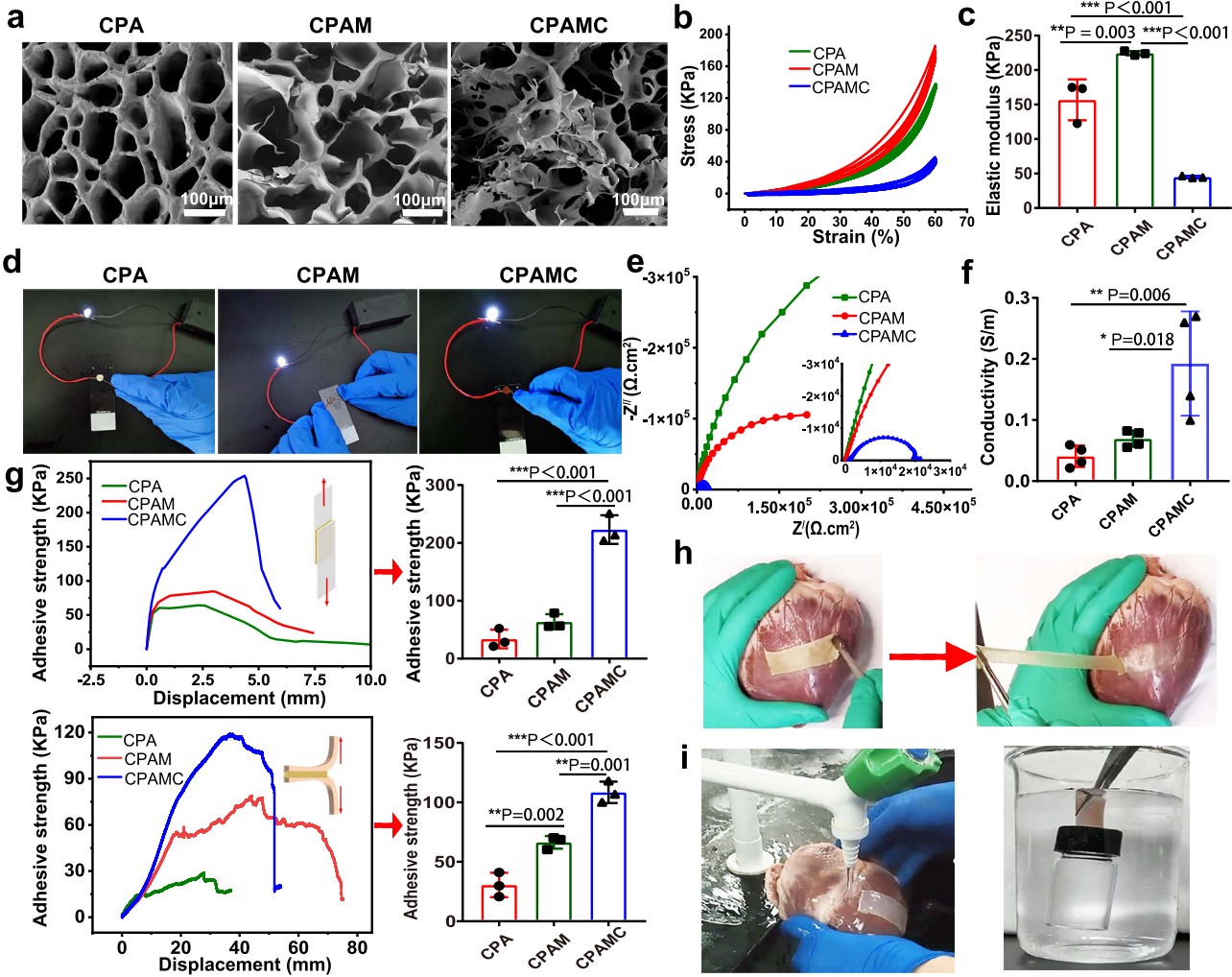

**Fig. 2 | The morphological, conductive, and mechanical characteristics of the adhesion hydrogels. a** SEM images of CPA hydrogel, CPAM hydrogel, and CPAMC hydrogel, scale bars: 100 μm. **b** The stress–strain curves of CPA hydrogel, CPAM hydrogel, and CPAMC hydrogel. **c** The elastic modulus of CPA hydrogel, CPAM hydrogel, and CPAMC hydrogel (error bar means the standard deviation, *$p < 0.05$, **$p < 0.01$, ***$p < 0.001$, $p$ value was generated by one-way analysis of variance (ANOVA), followed by Tukey's multiple-comparison post hoc test, $n = 3$ independent samples). **d** The luminance changes of the LED light bulb in CPA hydrogel, CPAM hydrogel, and CPAMC hydrogel. **e** Nyquist plots of CPA hydrogel, CPAM hydrogel, and CPAMC hydrogel. **f** Statistical analysis of conductivity in CPA hydrogel, CPAM hydrogel, and CPAMC hydrogel (error bar means the standard deviation, *$p < 0.05$, **$p < 0.01$, ***$p < 0.001$, $p$ value was generated by one-way analysis of variance (ANOVA), followed by Tukey's multiple-comparison post hoc test, $n = 4$ independent samples). **g** Adhesion properties of three hydrogels in dry (glass, upper column) and wet states (muscle tissue of pig heart, bottom) (error bar means the standard deviation, *$p < 0.05$, **$p < 0.01$, ***$p < 0.001$, $p$ value was generated by one-way analysis of variance (ANOVA), followed by Tukey's multiple-comparison post hoc test, $n = 3$ independent samples). **h** Images of CPAMC hydrogel adhering to pig heart. **i** Images of CPAMC hydrogel tightly adhered porcine heart under water flushing (left) and supporting a full-filled bottle under water (right). CAP, CAPM, CPAMC: C, aldehyde cellulose (CNC-CHO); P, polyethylenimine (PEI); A, acrylic acid; M, 3-sulfonic acid propyl methyl acrylic acid potassium (MASEP); C, caffeic acid (CA).

1680 cm⁻¹, indicating that the successful formation of dynamic imine bonds between CNC-CHO and PEI in hydrogel network. Compared CPA hydrogel, CPAM and CPAMC hydrogels showed a strong absorption peak at 1153.96 cm⁻¹ assigned to sulfonic group, which manifests that the MASEP was successfully introduced into the hydrogel. The absorption peak around 733.1 cm⁻¹ corresponding to the phenolic hydroxyl (Ph-OH) stretching of caffeic acid appeared in the CPAMC hydrogels, suggesting the successful introduction of catechol structure into the hydrogel polymeric network[32].

The microstructures of CPA, CPAM, and CPAMC hydrogels were observed by scanning electron microscope (SEM). As represented in Fig. 2a, all three hydrogels after freeze-drying exhibited typical porous microstructures. Compared with single-network CPA hydrogel, an entirely different morphology with numerous interconnected porous structures was observed in CPAM and CPAMC samples. These interconnected porous structures may prevent crack propagation and

facilitate the transfer of ions within hydrogels, which can enhance ions conductivity of hydrogel[33].

The mechanical properties of the hydrogel scaffold play a vital role in the function of ECP. An ideal scaffold for ECP should possess two mechanical properties. Firstly, an appropriate elastic modulus that is close to the natural heart tissue can serve for the synchronous contraction of cardiomyocytes in vitro[34], and secondly, an excellent fatigue resistance can support steady cardiac systole and diastole after it is transplanted onto the infarct heart in vivo. To this regard, we conducted a cycling compressive test on three different hydrogels. All samples did not get rupture after compressing to 60% strain. In addition, there is no significant change in the compression curve after repeated compression, indicating that the addition of CNC can prevent crack propagation and make the bonded hydrogel have good fatigue resistance (Fig. 2b). The elastic modulus and mechanical strength of hydrogel can be significantly improved by the interpenetrating

network formed by MASEP, which elastic modulus increased from $156.86 \pm 29.7$ kPa to $224.4 \pm 3.13$ kPa in a 60% compress strain after adding MASEP (Fig. 2c). However, the elastic modulus of CPAMC hydrogel reduced to $45 \pm 1.5$ KPa after introducing caffeic acid. The elastic modulus of all hydrogels is within the elastic modulus range of native myocardium[35], meeting the requirements of elasticity and fatigue resistance of scaffolds for cardiac tissue engineering.

To check the electrical conductivity in these hydrogels, we designed a complete circuit composed of a light-emitting diode (LED) to evaluate the electrical performance of various hydrogels. As shown in Fig. 2d, CPA hydrogel can also make the LED emit light, but the brightness is very weak. When adding MASEP, CPAM hydrogel can make the LED emit brighter light, indicating that the addition of ionic agent and the formation of interpenetrating networks significantly improves the conductivity of the hydrogel. In addition, we see that the CPAMC hydrogel with CA became more conductive, and the light LED gave off brighter light. The electrochemical impedance spectroscopy (EIS) and cyclic voltammetry (CV) was conducted to further evaluate the electrical properties of these three hydrogels. The EIS data were analyzed via Nyquist plots containing a semicircle at a high frequency and a straight line at a low frequency. Large diameter of the semicircle indicated high charge-transfer resistance attributed to the poor electrical conductivity of samples[12]. Based on EIS curves in Fig. 2e, it is obvious that the diameter of semicircle keeps decreasing with the addition of MASEP and caffeic acid, implying the increase of electrical conductivity. We also obtained the same trend of CV curves using the cyclic voltammeter method (Supplementary Fig. 4). It is notable from CV curves that CPAMC sample shows the largest area of hysteresis loop and the most obvious pair of redox peaks, indicating that CPAMC sample has the best conductivity, which is consistent with the results from EIS. In Fig. 2f, CPA hydrogel without MASEP shows a low conductivity of $0.041 \pm 0.017\,S\,m^{-1}$, CPAMC hydrogel with MASEP increased to $0.069 \pm 0.013\,S\,m^{-1}$. CPAMC hydrogel with MASEP and CA possesses an optimal conductivity of $0.193 \pm 0.085\,S\,m^{-1}$, which completely accords with the electrical signal transduction level of the native myocardium tissue ($\approx 0.1\,S\,m^{-1}$). The above analysis consistently proves that CPAMC hydrogel possesses an appropriate electrical conductivity matching with native cardiac tissue, enabling it great potential for adjusting electrophysiology in cardiac tissue.

## Adhesion performance of the CPAMC/PCA Janus hydrogel

Excellent adhesion performance is not doubt charming for MI repair, the firm adhesion between the cardiac tissue and conductive patch could provide suitable mechanical support, quickly achieve electrophysiological conjugation and integration, and facilitate cardiac function recovery. Hence, we further examined the adhesion properties of the Janus hydrogel in dry and wet state. The fresh porcine myocardium tissue was used to measure the peeling adhesion properties of the adhesive hydrogels in the wet state, and glass slide was used as a representative to measure the lap-shear adhesion properties of the hydrogels in the dry condition, so as to evaluate the adhesion properties of the three hydrogels. The maximal detachment stress of the hydrogel from the substrate was recorded as the adhesive strength. As demonstrated in Fig. 2g, with the addition of MASEP and CA, the lap shear strength and peel adhesion strength of CPAM and CPAMC hydrogels gradually increased in both wet and dry conditions. CPAMC hydrogels have the best adhesion performance, which is about 223.14 KPa in dry state and 108.41 KPa in wet state. By contrast, CPA and CPAM hydrogels presented an obvious lower shear adhesion strength both in dry state and wet state. These results substantiated our suppose for enhanced interfacial interactions. The improved tissue adhesion of the CPAMC hydrogel benefits from the enhancement from multiplex dynamic noncovalent interactions and physical entanglement of PEI and Poly (AA-co-MASEP) polymeric chains by introducing the ionization promoter MASEP and caffeic acid (CA) with catechol

groups. The adhesion performance of the CPAMC hydrogel to surfaces of different dry materials and wet bio-tissues was further evaluated in Fig. 2h, i and Supplementary Fig. 5. As shown in Fig. 2h, CPAMC hydrogel has excellent adhesion performance on pig heart, and the hydrogel can still firmly adhere to the heart after being stretched twice. CPAMC hydrogel can stick 50 g weight in water and do not fall off even under flushing by running water (Fig. 2i, Supplementary Movie 1, and Supplementary Movie 2). Also, the CPAMC hydrogel can firmly adhere to both hydrophilic and hydrophobic surfaces, such as rubber, glass, and plastic, and even support a weight of up to 1.5 kg (Supplementary Figs. 5b, e). Especially, the CPAMC hydrogel exhibited repeatable adhesiveness to a variety of substrates with considerable durability for more than 30 cycles of attachment/detachment tests (Supplementary Fig. 5a). Moreover, the CPAMC hydrogels also exhibited high adhesiveness to various wet organs, including heart, lung, kidney, spleen, liver, which is crucial for biomedical applications (Supplementary Fig. 5c). In addition, to further verified the role of CNC-CHO on the adhesion of the hydrogels, we check the adhesion performance of CPAMC hydrogels with or without containing CNC-CHO. It is found that the PAMC hydrogels without CNC-CHO could not lift a weight of 50 g stably under dry environment, while the CPAMC hydrogels containing CNC-CHO could easily lift a 50 g object even under water (Supplementary Fig. 6a). The lap-shear curves of the above two hydrogels also demonstrated that the adhesion strength of CPAMC hydrogel is significantly higher than that of the PAMC hydrogel (Supplementary Fig. 6b), which reveals that the sulfonated and aldehyde-functionalized CNC are beneficial for the adhesive capacity of the hydrogel, which is consistent with previously reported results[30]. More impressively, the CPAMC hydrogel can stick tightly to the surface for a considerable long term over 30 days and still exhibit relatively good stretching properties of the same CPAMC hydrogel bearing large deformation under stretching without breaking after 30 days of adhesion (Supplementary Fig. 5f). These results indicate that the CPAMC hydrogel has good adhesion to not only the tissue surfaces but also other various organic and inorganic material surfaces and can be stably fixed in the myocardial infarction area. Such robust adhesiveness was mainly ascribed to covalent or/and noncovalent reactions between hydrogel and a variety of substrates[17,18]. Specifically, the robust covalent crosslinking networks inside the hydrogel and between tissue surface and hydrogels and multiple strong interactions of substantial polar groups retained inside the hydrogel enabled a strong adhesion through hydrogen bonding, coordination bonding, cation–π interaction, and so on (Fig. 1 and Supplementary Fig. 5d).

## Tissue-like adhesion-stiffening and molecularly encoded adhesion-adaptive mechanism of the CPAMC/PCA Janus hydrogel

Strain-stiffening hardening behavior is a basic characteristic of biological tissue, which can protect itself from damage caused by large deformation[36–39](Fig. 3a). Mimicking biological stress–strain behavior using synthetic materials is fantastic, yet challenging for various promising applications, especially for tissue engineering. As far as we know, a synthetic polymeric system with tissue-like adhesion stiffening has not been reported. Herein, in our work, it is intriguing to found that the adhesion properties of CPAMC hydrogels also showed strain-stiffening behavior. We traced the adhesion performance of CPAMC hydrogels with a duration of 1 month (corresponding to per conventional treatment cycle for MI repair in animal model). It is interested to found that, with the increase of adhesion time, the mechanics of hydrogel increased, and the adhesive strength also increased. As shown in Fig. 3c–e, when the strength of hydrogel increased from 25.35 to 207.60 kPa, the adhesive strength increased from 251.14 to 16101.15 kPa, it can easily lift up to 1.5 Kg (Supplementary Fig. 5e and Supplementary Movie 3). Although the tensile properties of the hydrogel decreased with the increase of strength, it still maintained a 50% tensile

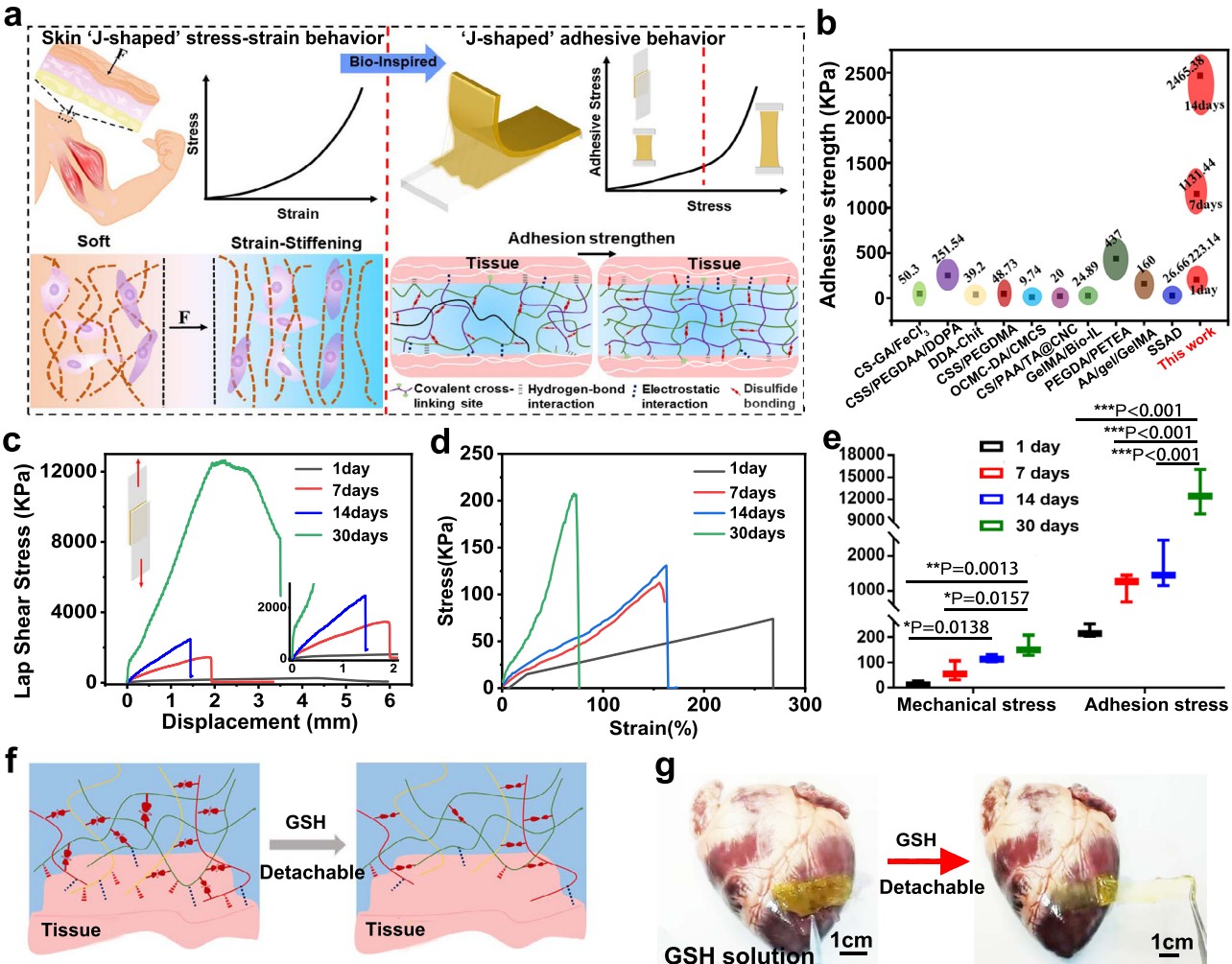

**Fig. 3 | Insight into developing CPAMC superglue mimicking from skin sclerotization process. a** Schematic illustration of strain-stiffening behavior of CPAMC hydrogel. **b** Comparison of adhesion strengths of our CPAMC in gluing glass with the literature values[13,33,54]. **c** Adhesive strength of CPAMC hydrogel in glass in air condition with different contact time. **d** Mechanical strength of CPAMC hydrogel in glass in air condition with different contact time. **e** Statistical analysis of adhesive strength and mechanical strength of CPAMC hydrogel in glass in air condition with different contact time (error bar means the standard deviation, *$p < 0.05$, **$p < 0.01$,

***$p < 0.001$, $p$ value was generated by one-way analysis of variance (ANOVA), followed by Tukey's multiple-comparison post hoc test, $n = 3$ independent samples). **f** Schematic diagram of dissociation of CPAMC hydrogel triggered by GSH. **g** Picture of dissociation of CPAMC hydrogel triggered by GSH. CAP, CPAM, CPAMC: C, aldehyde cellulose (CNC-CHO); P, polyethylenimine (PEI); A, acrylic acid; M, 3-sulfonic acid propyl methyl acrylic acid potassium (MASEP); C, caffeic acid (CA).

rate, in line with the systolic diastolic strain of the heart (Supplementary Fig. 5f). In addition, we compared the adhesion strength of CPAMC hydrogels with the reported adhesion strength, and found that our hydrogels had excellent adhesion far exceeds these reported wet adhesion values (Fig. 3b).

Similar to several advanced artificial systems mimicking biological strain-stiffening behavior underlying the mechanics from fibrous networks confined by close-packed cells of bio-tissues, it is probable that the adhesion-stiffening behavior of CPAMC hydrogels is due to the topographical structural and mechanical adaptability caused by dynamic responsive crosslinking hydrogel network. When adhered to wet tissues, the formation of the physically noncovalent and chemically covalent linkages continues to gradually evolve in the CPAMC hydrogel (Fig. 3a). Along with the progressively adjustment and enhancement of interpenetrating polymeric network, the embedded water was expelled and eventually, the adhesion strength hydrogels were prominently increased from the originally 251.14–16101.15 kPa after one month. An obviously positive correlation between the increase in the mechanical strength and adhesion stress can be found in Fig. 3e. Thus, we deduced that such redox responsive systems

became critical to enable an adaptive adherence of the resultant CPAMC hydrogel under a dynamic redox environment in vivo. In the current system, at one hand, the dynamic rearrangement induced by the physiologically responsive disulfide bond in the hydrogel could gradually extrude the residue water, which leads to the adhesion and mechanical property enhancement behavior after the hydrogel being adhered for a relatively long time. At the other hand, from our in vivo implantation result, the firmly adhesion of wet tissue after 4 weeks indicates that this dynamical reconstruction of interpenetrating network and adaptive mechanical stiffening within the responsive hydrogel to the MI microenvironment is also responsive for the enhanced adhesion capacity. The existence of redox-responsive disulfide bond performed an arresting function similar to the self-protective stress–strain effect of skin, muscle, and other tissues, which exhibited a dynamical reconstruction of interpenetrating network and adaptive mechanical stiffening within the hydrogel, and thus gradually extruding the residue water and achieving an enhanced adhesion ability to substrate surfaces. And this possible mechanism of this adhesion-stiffening phenomenon may be consistent with previously reported proposed mechanism[40]. Nevertheless, the molecularly

encoded adhesion-adaptive mechanism of this Janus hydrogel is needed to be further explored, but the misty mechanism does not abate and diminish the perfect performance and potential of our smart CPAMC/PCA Janus hydrogel as a promising bio-adhesive mimicking the biomechanical characteristic features, in particular with the smart and adaptive adhesion requirements for long-term repair of heart and other tissues accompanied by dynamically motion or mechanical movement.

## On-demand removability of CPAMC/PCA Janus hydrogel

Excellent adhesion capacity of CPAMC hydrogels ensures their tight and firm adhesion onto the targeted tissue surface, which is beneficial for restoring mechanically and electronically comfortable microenvironment and tissue repair performances. However, most of these adhesives are irreversible, and the good adhesion ability, in turn, may lead up to another puzzle that it's difficult to separate and remove from the adhesive site after the therapy on demand and may lead to secondary damage or new infection. Therefore, a smart adhesive hydrogel with reversible wet adhesion performance and, thus on-demand removable property could ideally meet the clinical needs. To this end, we chose redox-responsive BAC as a dynamic crosslinking agent. The on-demand dissolution capacity of smart CPAMC hydrogels was assessed via visual method. When oxidized glutathione was added, the redox-responsive disulfide bond of BAC in the adhesive layer hydrogel was broken and the adhesion ability was switched, which enable the hydrogel quickly separate and remove from the adhesive site (Fig. 3f, g and Supplementary Movie 4). This facile detachment of the adhesive hydrogel is nonirritating triggered by slight biologics, such as oxidized glutathione and biocompatible, moreover this removal process is feasible and easy-to-operate, making it possible for facile and on-demand removal of adhesive tissue scaffold avoiding additional damage.

## Comprehensive performance of CPAMC/PCA Janus hydrogel

As a functional cardiac patch, besides excellent adhesion and on-demand detachment ability, biocompatibility and proper porous microstructure, mechanical and electrical-conductivity properties are essential for further in vivo bioapplications. In order to prevent adverse tissue synechia, a Janus CPAMC/PCA hydrogel was fabricated through an anti-tissue-adhesion layer onto the topside of the adhesive bottom-layer hydrogel (Fig. 4a). Different from CPAMC hydrogels, PCA hydrogels lack catechol, amino, and aldehyde groups, but contain a large number of carboxyl groups and polyethylene glycol chains, so that PCA hydrogels can reduce adhesion and exhibit obvious anti-cell-adhesion. As showed in FT-IR, PCA hydrogels had stronger absorption peaks than CPAMC hydrogels at 920 and 1710 cm⁻¹, indicating that the carboxyl content of PCA hydrogels was higher than that of CPAMC group (Fig. 4b). Subsequently, we tested whether the conductivity of the hydrogel with anti-adhesion layer changed, and found that the addition of anti-adhesion layer had little effect on the conductivity of the hydrogel (Fig. 4c). Scanning electron microscopy results showed that the adhesion layer and the anti-adhesion layer could be closely connected (Fig. 4d). The adhesion layer had a relatively close pore structure, while the anti-adhesion layer had larger pores, which was more favorable to the close adhesion of the patch to the myocardial infarction site. This asymmetrical porous microstructure is demonstrated to effectively promote anti-adhesion and pro-healing for injury repair of some biological cavity wall[6]. It is expected that the Janus porous microstructures can further enhance the multifunctionality of our CPAMC/PCA Janus hydrogel with inhibiting pericardial adhesion and facilitating MI repair. We also checked the overall mechanical properties of CPAMC/PCA Janus hydrogels, and found that the elastic modulus and fatigue resistance of CPAMC/PCA hydrogels (49.53 ± 2.48 KPa) exhibit no obvious difference from that of CPAMC hydrogels (45 ± 1.5 KPa) (Supplementary Fig. 7). These results indicate that the thin PCA anti-adhesion hydrogel layer had almost no impact on the mechanics of CPAMC hydrogel, and the elastic modulus of CPAMC/PCA hydrogel is still within the range of natural myocardium, which meets the mechanical requirements of cardiac tissue engineering for elasticity. The entirely opposite adhesive performances of the CPAMC side and CPA side for the CPAMC/PCA Janus hydrogel are shown in Fig. 4e and Supplementary Movie 5 in the Supporting Information. It can be seen that the CPAMC side of the CPAMC/PCA Janus hydrogel displays a high adhesion to pork skin, but the PCA side of the CPAMC/PCA Janus hydrogel shows a negligible adhesion. CPAMC/PCA Janus hydrogel can be fixed on the heart stably for a long time due to the high adhesion of CPAMC hydrogel layer, which can effectively prevent the adhesion between the heart and chest wall, and avoid the single anti-adhesion physical barrier being easy to fall off, which cannot effectively achieve the purpose of anti-adhesion for a long time. Inflammation often leads to tissue adhesion after surgery. To test the anti-cell-adhesion ability, we tested the recruitment effect of hydrogel on macrophages by inoculating RAW264.7 cells into the anti-adhesion hydrogel layer. Compared with the dense RAW264.7 cells on culture plate (plate) and CPAMC hydrogel, there were only a few cells on PCA hydrogel, and the statistics showed that the RAW264.7 cells on PCA hydrogel were only 1/30 of that on culture plate and CPAMC hydrogel, indicating that PCA hydrogel did not recruit macrophages (Fig. 4f, g). These results indicate that the increased anti-adhesive layer hydrogel can reduce the aggregation of inflammatory cells and inflammatory factors, thus reducing the generation of fibrosis and preventing tissue adhesion. Compared with the previously reported anti-viscosity barrier, CPAMC adhesive hydrogel can make PCA anti-viscosity hydrogel firmly adhere to the heart for a long time, and avoid the occurrence of postoperative adhesion caused by the shedding of heart beating.

## Biological functions of CPAMC hydrogel in vitro

Cell viability of the cardiomyocytes (CMs) seeded on CPA and CPAM and CPAMC hydrogels were determined by live-dead staining and CCK-8 assay at day 7. Live-dead staining results revealed that CMs grew well on all three hydrogels, while the cell density on the CPAMC hydrogels were higher than that on the CPA or CPAM hydrogels (Fig. 5a and Supplementary Fig. 8b). CCK-8 results showed that CMs activity on CPAMC hydrogels was significantly higher than that on CPAM and CPA hydrogels (Supplementary Fig. 8a). CPAMC hydrogel can accelerate CMs adhesion and extension because the catechol structure of CA can facilitate cell adhesion through covalent and non-covalent interactions[41]. After being cultured for 7 days, F-actin staining was performed to observe the morphology characteristics of the CMs. The cells seeded on the hydrogels exhibited well extended, indicating that all the three hydrogels had good biocompatibility (Fig. 5b). In addition, a large number of sarcomere structures in CMs could be observed on CPAM and CPAMC hydrogels, indicating that CMs on these two hydrogels owned mature contraction function. These results may owe to the introduction of ionization promoter MASEP. The ionization promoter endows CPAM and CPAMC hydrogels with outstanding electrical conductivity, and consequently, the improved conductive microenvironment promotes the electrical coupling between CMs. Herein, the prepared CPAM and CPAMC hydrogels can promote the functionalization of CMs.

Conductive scaffolds have been proved to contribute to the maturation of CMs and to promote synchronous contraction of the myocardium. To further examine the effect of conductive microenvironment on CMs maturation, the maturation of CMs in CPA, CPAMC, and CPAMC hydrogels was assessed by staining with sarcomeric α-actinin, cardiac troponin T (CTNT), and connexin 43 (CX43). Hereinto, α-actinin is prime microfilament protein for myocardium contraction[42], while CTNT is the regulatory protein of muscle tissue contraction, which is located on the fine myofilament of contractile protein and plays an important regulatory role in muscle contraction

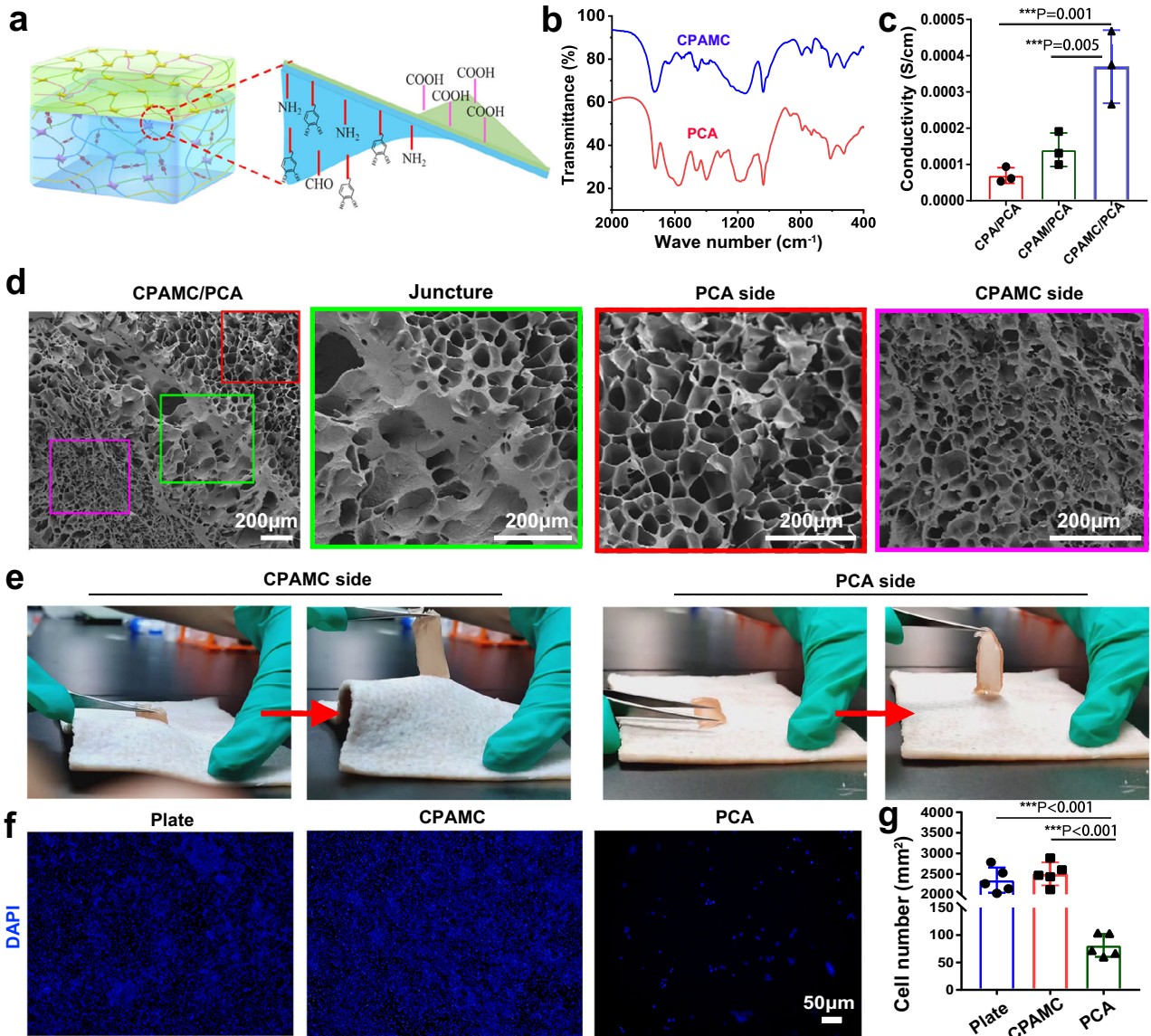

**Fig. 4 | Performances test of CPAMC/PCA Janus hydrogel. a** Schematic illustration of CPAMC/PCA Janus hydrogel. **b** The FT-IR of CPAMC hydrogel and PCA hydrogel. **c** The conductivity analysis of CPA/PCA, CPAM/PCA, and CPAMC/PCA Janus hydrogel (error bar means the standard deviation, *$p < 0.05$, **$p < 0.01$, ***$p < 0.001$, $p$ value was generated by one-way analysis of variance (ANOVA), followed by Tukey's multiple-comparison post hoc test, $n = 3$ independent samples). **d** The SEM of CPAMC/PCA hydrogel. **e** Adhesion analysis of CPAMC side and PCA side of CPAMC/PCA Janus hydrogel. **f** The recruitment results of RAW 264.7 cell in culture plate (plate), CPAMC hydrogel, and PCA hydrogel. **g** The statistic analysis of recruitment of RAW 264.7 cell in culture plate (plate), CPAMC hydrogel, and PCA hydrogel (error bar means the standard deviation, *$p < 0.05$, **$p < 0.01$, ***$p < 0.001$, $p$ value was generated by one-way analysis of variance (ANOVA), followed by Tukey's multiple-comparison post hoc test, $n = 5$ independent samples). CAP, CPAM, CPAMC: C, aldehyde cellulose (CNC-CHO); P, polyethylenimine (PEI); A, acrylic acid; M, 3-sulfonic acid propyl methyl acrylic acid potassium (MASEP); C, caffeic acid (CA). PCA: P, polyethylene glycol diacrylate (PEGDA); C, carboxylated cellulose (CNC-COOH); A, acrylic acid.

and relaxation[43]. CX43 is a well-known gap junction protein between CMs, which is closely involved in the conduction of electrical signals between cells[44]. As shown in Fig. 5c and Supplementary Fig. 9, primary CMs on CPA, CPAM, and CPAMC hydrogels showed significant expression of α-actinin and CTNT, indicating that all three hydrogels could promote CMs mature. However, compared with those on CPA hydrogel, the CMs on CPAM and CPAMC hydrogels showed more obvious sarcomere structure and higher expression of CX43, suggesting well-conductivity of the two hydrogels made the CMs with significant contractile function and gap junction intercellular communication. Moreover, the addition of MASEP and CA endow the CAPM and CPAMC hydrogel with favorable conductive microenvironment for the CMs maturation and intercellular connection, and is conducive to the well contraction function of CMs.

The Ca²⁺ transient of CMs is usually taken as an indirect indicator of action potential propagation. To further investigate the effect of the conductive microenvironment of CPAMC hydrogel on CMs systolic synchrony, the Ca²⁺ transient among CMs on different hydrogels were measured and three different points were randomly selected for analysis. Strong, high frequency and synchronous calcium signal changes among CMs on the CPAMC hydrogel were observed. In contrast, weak and asynchronous Ca²⁺ puffs among CMs occurred in CPA hydrogels (Fig. 5d–f and Supplementary Movie 6). These results indicated that CPAMC hydrogel with MASEP and CA could promote electric signal conduction and established an entire synchronization among distinct clusters of beating CMs without an external electrical impulse. On day 4 of culture, spontaneous contraction of CMs-seeded CPAMC hydrogel could be observed under microscope and larger contraction

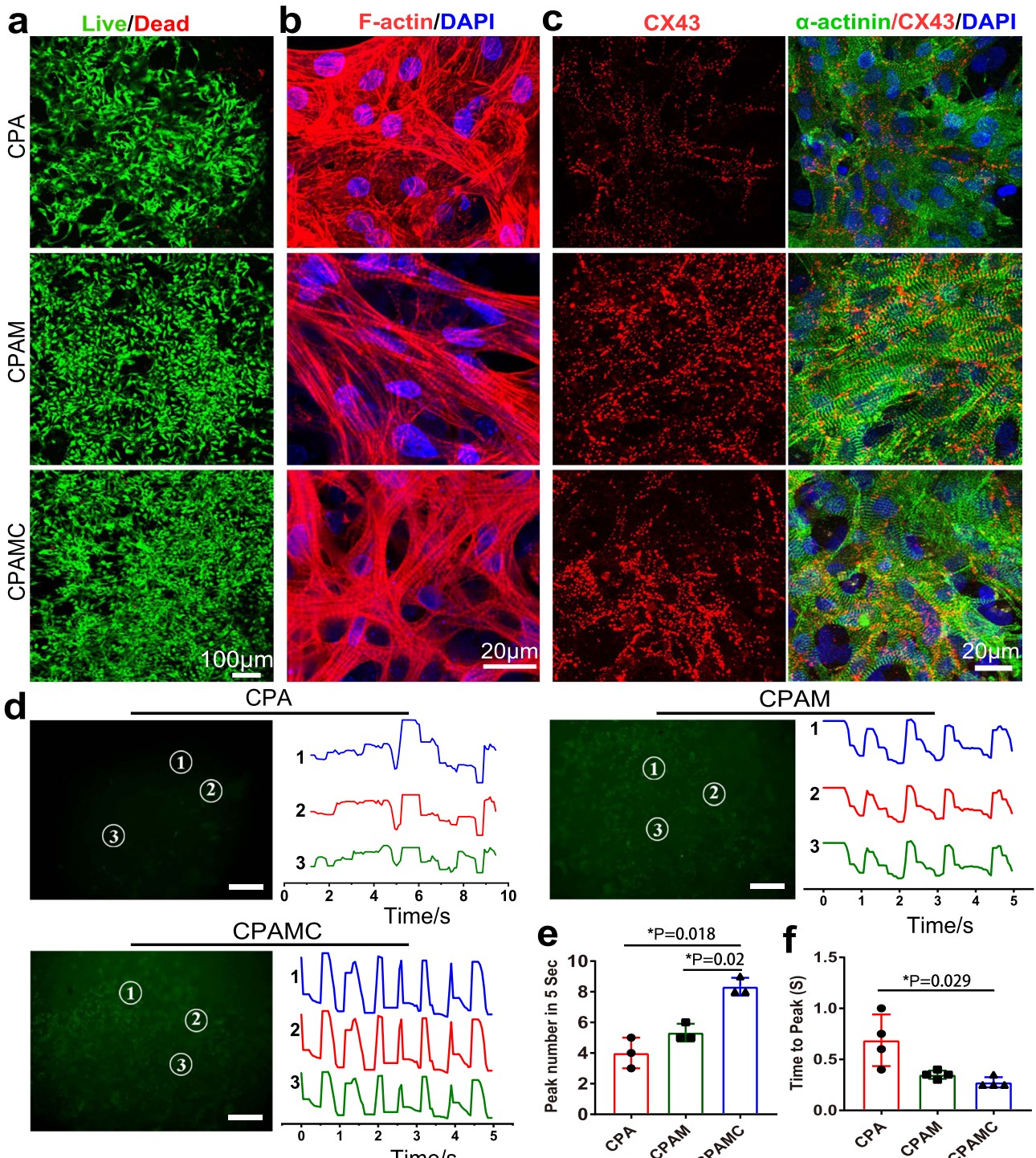

**Fig. 5 | The morphological and functional characteristics of CMs in different hydrogels. a** Live/dead staining of CMs on CPA hydrogel, CPAM hydrogel, and CPAMC hydrogel. **b** The F-actin stained of cytoskeleton of CMs on CPA hydrogel, CPAM hydrogel, and CPAMC hydrogel. **c** Expression of cardiac-specific proteins of α-actinin (green) and CX-43 (red) in the CMs on CPA hydrogel, CPAM hydrogel and CPAMC hydrogel. Scale bars: 20 μm. **d** Calcium transient of CMs on different hydrogels at day 7 of culture. Scale bars: 100 μm. **e** The Ca²⁺ transient propagation peak numbers in different hydrogels were calculated every 5 s based on the image of (**d**) (error bar means the standard deviation, *$p < 0.05$, **$p < 0.01$, ***$p < 0.001$,

$p$ value was generated by one-way analysis of variance (ANOVA), followed by Tukey's multiple-comparison post hoc test, $n = 3$ independent samples). **f** Ca²⁺ transient propagation time to peak in different hydrogels based on the image of (**d**) (error bar means the standard deviation, *$p < 0.05$, **$p < 0.01$, ***$p < 0.001$, $p$ value was generated by one-way analysis of variance (ANOVA), followed by Tukey's multiple-comparison post hoc test, $n = 4$ independent samples). CAP, CPAM, CPAMC: C, aldehyde cellulose (CNC-CHO); P, polyethylenimine (PEI); A, acrylic acid; M, 3-sulfonic acid propyl methyl acrylic acid potassium (MASEP); C, caffeic acid (CA).

amplitude could be detected on day 6 (Supplementary Fig. 10 and Supplementary Movie 7). On day 7 of culture, the spontaneous beating of the CMs-seeded CPAMC hydrogel became very strong and could be observed even in naked eyes (Supplementary Movie 7). This high-frequency and monolithic beating behavior mainly attributed to the formation of tightly connected mature CMs on the CPAMC hydrogel. The existence of icon-conductive hydrogel network with MASEP and CA endows CPAMC hydrogel with favorable electrical conductivity, which is conducive to the transmission of electrical signals between CMs and the establishment of synchronous contraction. We supposed that our developed conductive CPAMC hydrogel could be taken as ECP directly, which could bridge the electrical conduction between scar tissue and normal cardiac tissue after MI and facilitate the restore of normal electrical transmission.

After MI, a large number of ROS will generate, which induce continuous damage to CMs, and result in apoptosis and necrosis of CMs, and ultimately lead to the loss of cardiac function[15]. Therefore, it is no doubt that the cardiac patch scaffold with antioxidant and ROS scavenging capacity is attractive and effective for the repair of myocardial infarction. Owing to the existence of abounding antioxidant components in these hydrogels, such as disulfide bond in BAC, aldehyde group in CNC, catechol group in CA and so on, the resulting Janus hydrogel is endowed with a glorious oxidation resistance to restore the cardiac functions and facilitate MI repair effect. Hence, to confirm this antioxidant ability and potential benefits, we measured the antioxidant activities of CPA, CPAM, and CPAMC hydrogel by multiple assays using an oxidative stress cell model induced by hydrogen peroxide ($H_2O_2$).

After 12 h incubation with 500 μM $H_2O_2$, the survival of CMs on different hydrogels was evaluated by live-dead staining. As shown in Fig. 6a, $H_2O_2$ treatment dramatically decreased the survival capacity of CMs due to oxidative damages. Compared with $H_2O_2$ and CPA groups, CPAM and CPAMC hydrogels could significantly protect CMs from the oxidative damages and improve the CMs activity. Similar results were also detected using CCK-8 assay, in which the cell viability of CMs on CPAM and CPAMC hydrogels was much higher than that of CMs cultured on plate or CPA hydrogel (Fig. 6c). Subsequently, to verify the ROS scavenging activity of CPAMC hydrogels, 2′,7′-dichlorofluorescin diacetate (DCFH-DA) staining was used to assess intracellular ROS levels. As shown in Fig. 6a, lots of strong green fluorescence (DCFH-DA) appeared around the blue nuclei (4′,6′-diamidino-2-phenylindole DAPI) in CMs in pure $H_2O_2$ treatment group, indicating a high level of ROS. With the increase of the antioxidant components in hydrogels, the intensity of DCFH-DA fluorescence in CMs decreased, and the CMs seeded on CPAMC hydrogel had the weakest fluorescence. These results verified that CPAMC hydrogel possessed the excellent ROS scavenging ability. Finally, SOD activity and MDA content in CMs were measured respectively to further verify the cellular oxidation resistance in different hydrogels. SOD plays a key role in oxidation/antioxidant balance, which can clear superoxide anions and free radical, and repair damaged cells timely. As a product of lipid oxidation, MDA can indirectly reflect the cellular damage caused by oxidative stress[45]. As shown in Fig. 6d, compared with those in $H_2O_2$ and CPA groups, CMs on CPAM and CPAMC hydrogels exhibited higher level of SOD activities for the elimination of numerous ROS. The increase of cellular SOD activity was attributed to the antioxidant properties of CPAM and CPAMC hydrogels. Furthermore, MDA content in CMs seeded on CPAMC hydrogel was significantly lower than that in CMs seeded on $H_2O_2$ and CPA groups (Fig. 6e). The results indicated that CPAMC hydrogel could effectively remove ROS and markedly reduce the oxidative damages and improve cells viability under oxidative stress microenvironment. The redox-responsive hydrogel network and the introduction of the catechol structure enable CPAMC hydrogel with excellent antioxidation performance and ROS scavenging ability[14]. The

CPAMC hydrogel with CA was supposed to elevate the survival capability of MI myocardium through ROS scavenging in vivo.

After MI, macrophages are rapidly activated from M0-type to M1-type, which secrete harmful inflammatory cytokines and further lead to myocardial injury[46]. It is very important to polarize M1-type macrophages into M2 types, which can secrete anti-inflammatory factors to inhibit inflammatory injury and promote the repair of myocardial injury. Inspired by the anti-inflammatory effect of natural polyphenol caffeic acid and sulfonated polymer of MASEP, the developed CPAMC hydrogel was supposed to have the ability on macrophage polarization in the MI inflammatory microenvironment. Herein, the RAW264.7 macrophages were used to assess the polarized ability of the CPAMC hydrogel. RAW264.7 cells were inoculated on hydrogels firstly, and then 100 ng/mL lipopolysaccharide (LPS) was added to produce an inflammatory environment. As shown in Fig. 6b, cells in pure LPS group mainly expressed M1 macrophage proteins (iNOS), while cells in CAPM and CPAMC groups mainly expressed M2 macrophage proteins (TGF-β), especially almost only TGF-β was expressed in CPAMC hydrogel. This is because the sulfate groups in CPAM and CPAMC hydrogels have the ability to promote the polarization of M1 to M2, and the presence of CA in CPAMC further improves the anti-inflammatory ability of CPAMC hydrogels[19,47]. These results suggest that CPAMC hydrogel can enhance the M2-type polarization of macrophages and is beneficial of MI repair.

Taken together, the oxidative stress and inflammatory activation in early stage of MI are key mechanisms of ischemic myocardial injury. It is, therefore, foreseeable that CPAMC hydrogel transplantation in the infarcted myocardium helps to prevent ventricular remodeling after MI and improves cardiac function considering its well performance of rapidly scavenging a large number of ROS.

## Repair effects of CPAMC/PCA ECP in rat MI models

Although a large number of cardiac patches had been developed for injury repair after MI, most of these patches require invasive fixation and often cause severe tissue synechia. It is a great challenge to develop a multifunctional cardiac patch that can simultaneously achieve MI repair and postoperative tissue anti-synechia. Considering the physicochemical microenvironment of Janus hydrogel in vitro can promote the maturation and functionalization of CMs, and can reduce the oxidative damage of CMs, and promote the polarization of reparative inflammatory cells, we further studied in vivo whether this microenvironment can promote the repair of myocardial infarction injury without the participation of exogenous cells. After one week of post-operation, CPAM, CPAMC, and CPAMC/PCA ECPs were transplanted onto the infarcted region in MI rats for 4 weeks respectively (Fig. 7b). Except from CPAMC/PCA Janus hydrogel transplantation group, additional rat studies were performed to compare the treatment effects of different layers from a single-layer CPAMC patch without anti-synechia PCA layer (CPAMC group) and a single-layer ionic-conductive CPAM hydrogel patch without antiadhesive PCA layer and caffeic acid (CPAM group).

From the echocardiographic images in Fig. 7a, we can see that bare contraction occurred in the left ventricular anterior wall in the MI group and smaller contraction waves in the CPAM group, while obvious contractile activity of the left ventricle anterior wall was observed in the CPAMC and CPAMC/PCA ECP transplantation group after transplantation for 4 weeks. Typical echocardiography parameters, ejection fraction (EF), fractional shortening (FS), left ventricular internal dimension in systole (LVIDs), and left ventricular internal dimension in diastole (LVIDd) were analyzed to evaluate cardiac function in rats[48]. Both EF and FS were significantly enhanced in the CPAM ECP group, CPAMC ECP group and the CPAMC/PCA ECP group compared to those in the MI group. Among them, the CPAMC and CAPMC/PCA ECP transplantation group possessed high EF and FS

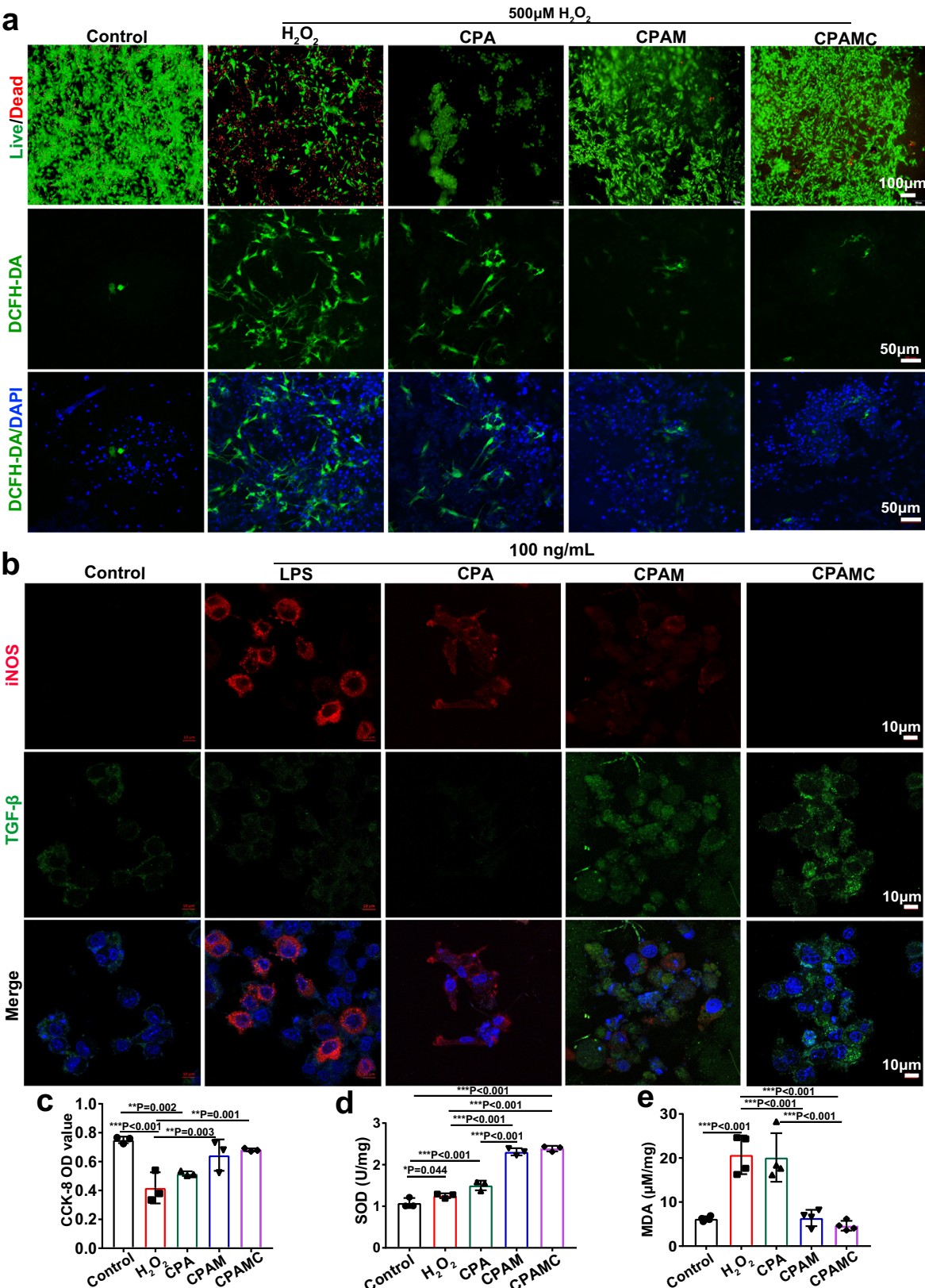

values (Fig. 7c–f). Consistent with the promotion in EF and FS, LVIDd and LVIDs are efficiently prevented on negative LV dilation in CPAMC/PCA ECP (Supplementary Fig. 11). These results suggest that CPAM, CPAMC and CAPMC/PCA ECP with good electrical conductivity, anti-inflammatory, and antioxidant properties could restore cardiac function after MI, especially CPAMC and CAPMC/PCA ECP with MASEP

and CA. Additionally, the excellent tissue adhesion of hydrogel scaffold also is beneficial for the minimization of secondary harm during transplantation and further safeguards of the injured cardiac function during and after healing.

After transplantation for 4 weeks, these patches were observed to well adhere to the surface of heart, validating their satisfactory wet

**Fig. 6 | Anti-oxidation capacity and anti-inflammatory effect of hydrogels.**
**a** Live-dead staining of CMs on CPA hydrogel, CPAM hydrogel, and CPAMC
hydrogel after 500 µM H$_2$O$_2$ treatment (above). The detection of ROS level in CMs
on CPA hydrogel, CPAM hydrogel, and CPAMC hydrogel after 500 µM H$_2$O$_2$
treatment (bottom). **b** Polarization of RAW264.7 cells on different hydrogels after
24 hours of LPS treatment at 100 ng/mL. **c** CMs activity (*n* = 3 independent sam-
ples), **d** SOD level (*n* = 3 independent samples), and **e** MDA level (*n* = 4 independent

samples) of CMs on different hydrogels after 500 µM H$_2$O$_2$ treatment (error bar
means the standard deviation, *$p$ < 0.05, **$p$ < 0.01, ***$p$ < 0.001, $p$ value was gen-
erated by one-way analysis of variance (ANOVA), followed by Tukey's multiple-
comparison post hoc test, *n* = 4 independent samples). CAP, CPAM, CPAMC: C,
aldehyde cellulose (CNC-CHO); P, polyethylenimine (PEI); A, acrylic acid; M,
3-sulfonic acid propyl methyl acrylic acid potassium (MASEP); C, caffeic acid (CA).

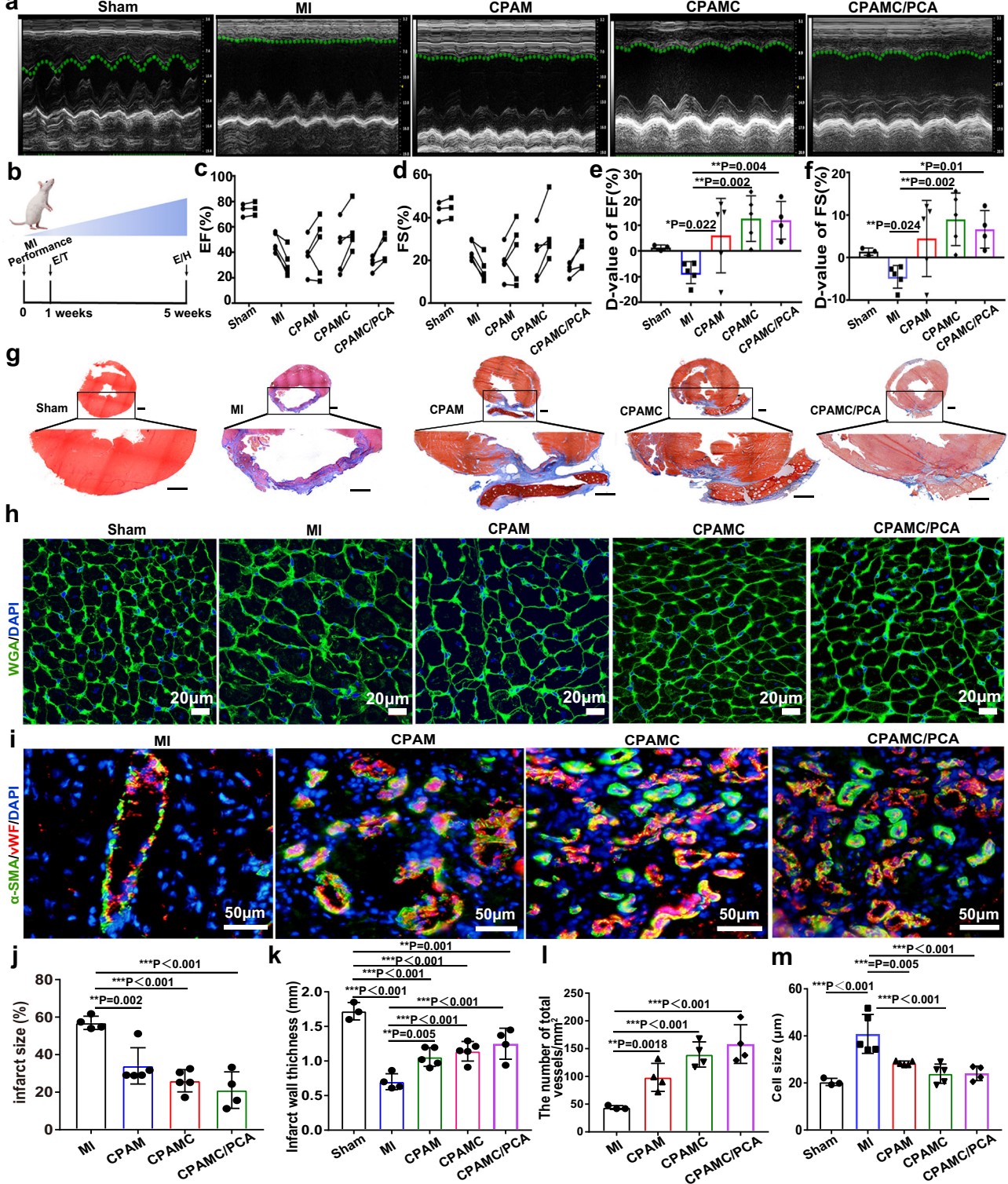

**Fig. 7 | Repair effect of different hydrogels in myocardial infarction rats. a** Left ventricular echocardiography in different treatment groups after patch transplantation for 4 weeks. **b** Experimental design. 1 week after MI performance, rats with FS lower than 30% were selected and performed ECP transplantation (T). The effects of the ECP were analyzed by echocardiography (E) and histology (H) at 4 weeks after transplantation. **c–f** Typical echocardiographic parameters of the left ventricular function for different groups (D-value: Difference value) ($n$ = animals: Sham group, $n$ = 3; MI group, $n$ = 5; CPAM group, $n$ = 5; CPAMC group, $n$ = 5; CPAMC/PCA group, $n$ = 4). **g** Masson trichrome staining displayed the fibrous tissue (blue) and myocardium (red) of sections of hearts from rats in different groups. Scale bars: 1 mm. **h** Wheat-germ agglutinin (WGA) staining showed myocyte boundaries in the border area, scale bar: 20 μm. **i** vWF immunostaining (red) and α-SMA immunostaining (green) within infarcted area in different groups, scale bar: 50 μm. **j, k** Statistical analysis of infarct size and infarct wall thickness of the infarcted heart in different group based on Masson trichrome staining ($n$ = animals: Sham group, $n$ = 3; MI group, $n$ = 4; CPAM group, $n$ = 5; CPAMC group, $n$ = 5; CPAMC/PCA group, $n$ = 4). **l** The microvessel densities within infarcted region in different groups based on vWF/α-SMA immunostaining staining ($n$ = animals: MI group, $n$ = 3; CPAM group, $n$ = 4; CPAMC group, $n$ = 4; CPAMC/PCA group, $n$ = 4). **m** Statistical analysis of cell size of myocytes based on WGA staining. ($n$ = animals: Sham group, $n$ = 3; MI group, $n$ = 5; CPAM group, $n$ = 5; CPAMC group, $n$ = 5; CPAMC/PCA group, $n$ = 4). Error bar means the standard deviation, *$p < 0.05$, **$p < 0.01$, ***$p < 0.001$, $p$ value was generated by one-way analysis of variance (ANOVA), followed by Tukey's multiple-comparison post hoc test. CPAM, CPAMC: C, aldehyde cellulose (CNC-CHO); P, polyethylenimine (PEI); A, acrylic acid; M, 3-sulfonic acid propyl methyl acrylic acid potassium (MASEP); C, caffeic acid (CA). PCA: P, polyethylene glycol diacrylate (PEGDA); C, carboxylated cellulose (CNC-COOH); A, acrylic acid.

tissue adhesion and stability. The cardiac tissues from Sham group, MI group, CPAM ECP group, CPAMC, and CPAMC/PCA ECP group were analyzed using Masson trichrome staining method, respectively. The results from Fig. 7g revealed that the fibrous tissue (blue) was formed in all MI models, in which the MI group showed the most severity of fibrosis. Comparatively, the CPAM group, CPAMC group, and CPAMC/PCA group presented decreased fibrosis. Notably, the CPAMC/PCA group demonstrates few fibrosis analogous to the Sham group. What is more, an adverse cardiomegaly is observed in both MI group and CPAM group but not in CPAMC and CPAMC/PCA group. Further, from the quantified analysis, the smallest infarct size (Fig. 7j) and the thickest LV wall (Fig. 7k) are observed in CPAMC/PCA-treated hearts compared to MI group. Compared with the MI group, CPAM group, CPAMC group, and CPAMC/PCA group have well-tissue-matched elasticity and conductivity, which could provide good mechanical support and electrical conduction to the infarcted myocardial tissue, and thus prevent ventricular enlargement and arrhythmia. In addition, CPAMC and CPAMC/PCA hydrogel with CA has excellent antioxidation capacity, which can effectively scavenge ROS in tissues and prevent the damage of CMs and the expansion of fibrosis. The anti-adhesion layer in CPAMC/PCA Janus hydrogels can reduce exogenous inflammatory injury by reducing the accumulation of exogenous inflammatory cells and inflammatory factors.

## CPAMC/PCA ECP could reduce inflammation and establish a regenerative microenvironment in the infarcted region

Inflammation plays an important role in myocardial repair, in which MI macrophages secrete a large number of pro-inflammatory factors to aggravate myocardial damages, while M2 macrophages secrete anti-inflammatory factors to promote myocardial repair[49]. In vitro studies showed that the hydrogel prepared by us could promote the polarization of M1 to M2-type macrophages. Herein, we further detected the recruitment of M1 and M2-type macrophages in myocardial infarction area. The CD86 positive M1 (red) and CD206 positive M2 (green) macrophages were immunostained respectively and demonstrated in Supplementary Fig. 12, only CD86 positive cells appeared in MI group, while both CD86 positive cells and CD206 positive cells coexisted in CPAM group. The decreased CD86 positive cells and the increased CD206 positive cells appeared in CPAMC group and CPAMC/PCA group. These suggest that the addition of MASEP in hydrogel could partially promote the polarization of M1-type macrophages to M2-type, but when MASEP and CA were present in CPAMC and CPAMC/PCA hydrogels, most M1-type macrophages could be polarized into M2-type, thus reducing the inflammatory damage in the myocardial infarction region and facilitating myocardial repair. More importantly, the presentation of anti-sticky layer hinders the invasion of foreign inflammatory cells and inflammatory factors, and the remarkable tissue adherence of the hydrogel scaffold significantly reduces secondary injury after transplantation as well as inflammation brought on by injury as well.

ROS is the main factor leading to apoptosis and necrosis of myocardial cells after myocardial infarction. To further evaluate the protective effect of CPAMC/PCA Janus hydrogel in myocardial infarction, cardiac sections were immunofluorescent stained with cardiac marker α-actinin after ECPs transplantation for 4 weeks. As showed in Supplementary Fig. 13, obvious α-actinin proteins were determined in the MI region in CPAMC and CPAMC/PCA group, but few α-actinin proteins appeared in the MI region in MI group. Quantitative data showed that the α-actinin positive ratio in the CAPMC/PCA group was similar to that in the CPAMC group, but much higher than that in the MI group. These results suggest that CA endowed CPAMC and CPAMC/PCA Janus hydrogels with outstanding antioxidant capacity, effectively inhibiting apoptosis or necrosis of CMs, and most surviving CMs were conducive to maintaining cardiac function after MI and effectively preventing ventricular remodeling.

## CPAMC/PCA Janus hydrogel could promote myocardial repair by enhancing angiogenesis

The recovery of blood supply in the infarct area can rescue the residual myocardial cells, provide sufficient nutrition for the tissues and restore cardiac function[50]. To evaluate the angiogenesis of the infarct area, anti-vWF antibodies were used to label the microvessels, and anti-vWF/anti-α-SMA antibodies were used to label the arterioles. As shown in Fig. 7i, both the microvessels (vWF⁺) and the arterioles (vWF⁺/α-SMA⁺) were significantly denser in the CPAMC and CPAMC/PCA group than those in the MI and CPAM groups. Complete and clear arteriole structures were observed in CPAMC and CPAMC/PCA groups, while less arteriole structure was detected in the MI group. Statistical analysis revealed that the number of blood vessels in the CPAMC hydrogel group was four times higher than that in the MI group (Fig. 7l). During the repair process, CPAMC hydrogel with MASEP and CA could accelerate the polarization of M1-type macrophages into M2-type macrophages, and M2-type macrophages can secrete a large number of cytokines promoting angiogenesis to promote the formation of blood vessels[51]. In addition, the conductive CPAMC and CPAMC/PCA also induced the vasculation, which facilitate nutrient delivery and cardiac function rebuilding in the infarction region.

After myocardial infarction, decreased cardiac function can lead to decompensated hypertrophy of cardiomyocytes, leading to ventricular remodeling. The reduction of M1 macrophages and ROS in the infarct area and the increasing of vascular supply can reduce the damaged CMs, restore the cardiac function and prevent the occurrence of ventricular remodeling, as well as inhibit hypertrophy of cardiomyocytes. Thus, we used wheat-germ agglutinin (WGA) staining to analyze the cardiac hypertrophy inhibition of our developed CPAMC/PCA Janus hydrogel (Fig. 7h). After 4 weeks post-MI, the CMs cross-sectional areas were significantly increased in the MI and CPAM groups, indicating obvious ventricular hypertrophy. On the contrary, the CMs cross-sectional areas in CPAMC hydrogel group and CPAMC/PCA Janus hydrogel group were significantly smaller than that in MI group. These indicated that both

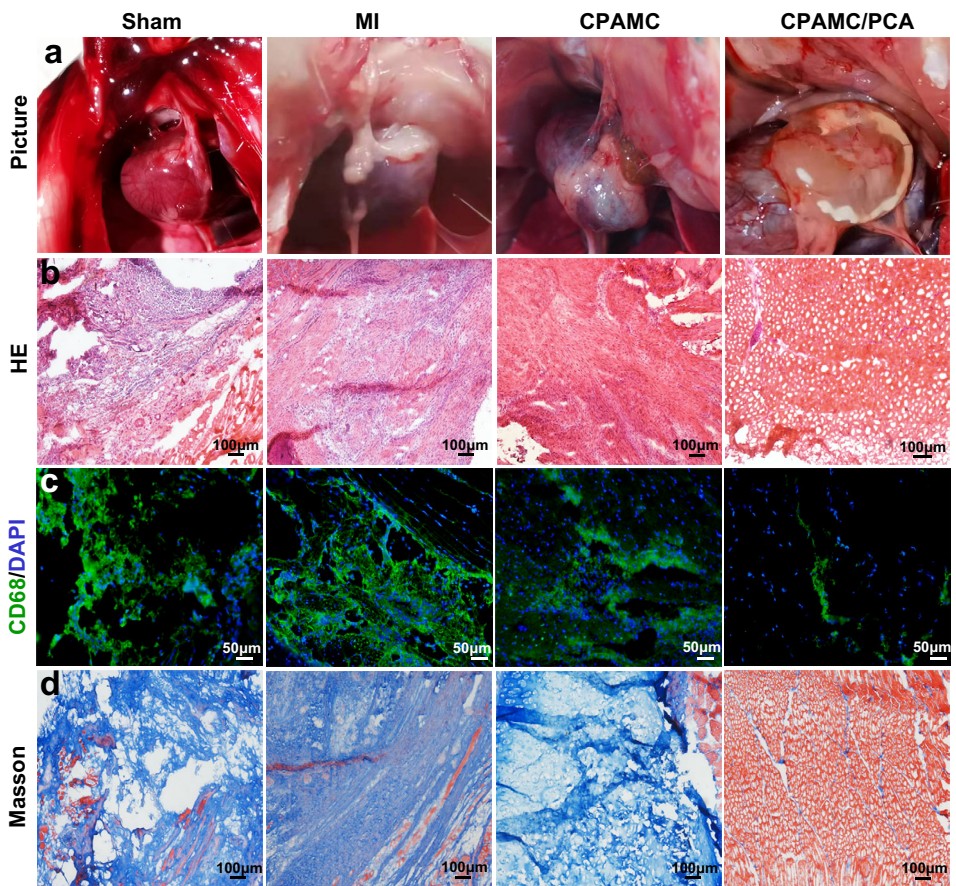

**Fig. 8 | Evaluation of anti-adhesion function of CPAMC/PCA hydrogel in MI rat model. a** Representation of heart adhesion to chest wall. **b, c** HE staining and immunofluorescence were used to evaluate the inflammatory status of chest wall tissue. **d** The fibrosis was assessed by Masson trichrome staining the chest wall tissue. CPAMC: C, aldehyde cellulose (CNC-CHO); P, polyethylenimine (PEI); A, acrylic acid; M, 3-sulfonic acid propyl methyl acrylic acid potassium (MASEP); C, caffeic acid (CA). PCA: P, polyethylene glycol diacrylate (PEGDA); C, carboxylated cellulose (CNC-COOH); A, acrylic acid.

CPAMC ECPs and CPAMC/PCA Janus ECPs transplantation could significantly reduce the hypertrophy of cardiomyocytes, and effectively restore cardiac function (Fig. 7m), which is consistent with echocardiographic data. These results suggest that CPAMC hydrogels and CPAMC/PCA Janus hydrogels could significantly reduce ventricular remodeling and prevent cardiac rupture after MI. The CPAMC hydrogel and CPAMC/PCA Janus hydrogel can reduce the oxidative damage and inflammatory damage of CMs in the infarct area and facilitate blood vessel regeneration. Also, the well electron-conductivity of CPAMC hydrogel and CPAMC/PCA Janus hydrogel reconstruct the electrical conduction between infarct area and normal cardiac area. Furthermore, the PCA anti-adhesion layer of CPAM/PCA can also prevent the recruitment of external inflammatory factors and inflammatory cells, and reduce exogenous inflammatory injury. The minimized oxidative and inflammatory damage and enhanced electric-coupling eventually reduce the occurrence of myocardial hypertrophy and accelerate the recovery of cardiac function.

**Synechia resistance of Janus CPAMC/PCA hydrogel in vivo**

Fibrous synechia always happens between the cardiac tissue and surrounding tissue after heart surgery, injury, and myocardial infarct. Fibrous synechia formation impedes cardiac function, seriously complicates thoracotomy by hindering visibility during operation and increasing the risk of death and morbidity[52,53]. To verify the anti-synechia performance of CPAMC/PCA Janus hydrogel ECP in vivo, we reopen the thoracic cavity after transplantation 4 weeks, and evaluated the anti-synechia effects of the patches. According to the gross

morphology of postoperative synechia shown in Fig. 8a, the Sham group, MI group, CPAMC hydrogel groups displayed varying degrees of tissue synechia. In particular, myocardial infarction triggered the most serious synechia formation, and the heart tissue was completely adhered with atop chest wall in MI group. This synechia formation might lead to undesirable aggregation of abundant inflammatory cytokine on the surface of heart and even trigger other complications. It is a stark contrast that our developed CPAMC/PCA Janus hydrogel patch was firmly adhered to the heart on the bottom side, and did not cause any thoracic synechia formation on the top side after 4-week implantation. This is attributed to the fact that the molecular chain rearrangement of the adhesive layer hydrogel caused by the BAC dynamic cross-linking system can enable the hydrogel to adaptively regulate adhesion under physiological conditions, while the PCA anti-adhesive layer reduces the aggregation of inflammatory cells and adhesion molecules. A small amount of synechia appeared in the region where the Janus patch uncovered, which further indicates the overwhelming anti-synechia effect of the Janus hydrogel. To further evaluate the anti-synechia and repair property of the CPAMC/PCA Janus hydrogel, cardiac tissue and chest wall samples from each group were then histologically evaluated. HE staining showed that thoracic wall tissues appeared a large number of inflammatory cell recruitment in the sham operation group, MI group, and CPAMC hydrogel group, indicating that these rats were in a serious inflammatory environment for a long time after surgery (Fig. 8b). At the same time, lots of CD68-positive inflammatory cells were observed in the chest wall tissue under immunofluorescence staining in the Sham group,

MI group and CPAMC hydrogel group, while these inflammatory cells were significantly reduced in the CPAMC/PCA Janus hydrogel group (Fig. 8c).

Inflammation usually leads to fibrotic repairs as well as fibrous adhesions. Masson trichrome staining showed that the thoracic wall tissues displayed a large amount of blue collagen tissues in the Sham group, MI group, and CPAMC hydrogel group, while displayed less blue collagen tissues and abundant red muscle tissues in the CPAMC/PCA hydrogel group (Fig. 8d). The above results indicating that the PCA hydrogel layer in the CPAMC/PCA Janus hydrogel exhibited a good anti-inflammation and anti-tissue synechia ability. In addition, HE staining and CD68 immunofluorescence staining were also used to evaluate the inflammatory situation on CPAMC hydrogel-attached and CPAMC/PCA hydrogel-attached cardiac tissues, and it was found that there was a mass of inflammatory infiltrations around CPAMC hydrogels, while no obvious inflammatory infiltrations around CPAMC/PCA hydrogels (Supplementary Fig. 14). The inner adhesion layer of CPAMC/PCA Janus hydrogel can facilitate the polarization of M1-type to M2-type macrophages in the infarct area, thus promoting the regeneration of blood vessels and tissue repair. Meanwhile, the PCA anti-adhesion layer offer a physically protective screen that obstruct the recruitment of inflammatory factors and inflammatory cells from surrounding tissues. This immune "walls-off" anti-adhesion layer prevented further damage of infarcted myocardium tissue by exogenous inflammatory factors. These results indicate that without the paracrine effect of exogenous cells, Janus hydrogel can provide excellent physicochemical microenvironment, reconstruct the electrical conduction in the myocardial infarction area, protect CMs from oxidative damage and inflammatory damage, promote the regeneration of blood vessels in the myocardial infarction area, and avoid the secondary damage and tissue adhesion caused by traditional patch transplantation, so as to effectively promote the repair of damage after myocardial infarction.

## Discussion

In this study, a multifunctional conductive hydrogel cardiac patch with Janus asymmetrical adhesion and on-demand removability was constructed for the first time to repair myocardial infarction and prevent post-MI tissue synechia and secondary trauma after suturing and removability. The adhesive layer of the hydrogel has strong wet and dry adhesion capacity, and the adhesive strength remarkably increases along with the adhesion time, which has an interesting adhesion-adaptive stiffening effect similar to the tissue-like stress–strain effect. The prepared adhesive hydrogel not only enables quickly adhere onto the wet and beating heart averting suture and other invasive fixation approaches, but also enable on-demand stimuli-triggered detachment for efficient myocardial infarction repair. In vitro and in vivo experiments showed that ion-conductive CPAMC/PCA Janus hydrogel can effectively reduce ROS and inflammation in myocardial infarction area, prevent the cascade of adverse cell signals and reduce ROS induced oxidative stress injury after myocardial infarction. In addition, CPAMC/PCA Janus hydrogel can reconstruct the electrical signal transmission of myocardium and promote tissue vascularization, inhibit the fibrosis and ventricular remodeling in myocardial infarction area, and enhance the cardiac function of infarcted rats. At the other hand, the anti-adhesion top layer of the Janus hydrogel can reduce the recruitment of macrophages and prevent the adhesion between postoperative patch and chest wall, which in turn benefits to the repair of cardiac function. The adhesive layer of the Janus hydrogel can be fixed to the target tissue for a long time in a non-invasive way, avoiding the disadvantage of easy shedding of the common anti-adhesion barrier and effectively preventing the occurrence of tissue adhesion after most surgeries. This study provides a versatile and efficient strategy for the design and construction of functional cardiac patch with synchronous barrier against postoperative tissue adhesion. This work also provides a handy and robust method to develop smart functional adhesive with on-demand removal and adhesion-adaptive stiffening effect similar to the stress–strain effect of biological tissues.

## Methods

### Materials

Acrylic acid (AA), 3-propyl methacrylate potassium (MASEP), sodium periodate ($NaIO_4$), Polyethylenimine (PEI), N, N'-bis(acryloyl) cystamine (BAC) and caffeic acid (CA) were purchased from Macklin (China). Alexa Fluor-568 donkey anti-rabbit IgG (H&L) and Alexa Fluor-488 donkey anti-mouse IgG (H&L) and F-actin dye were from Invitrogen. The Cell Counting Kit-8 (CCK-8) was supplied by Dojindo Molecular Technologies (Japan). The primary antibodies against α-actinin (ab9465), connexin 43 (CX-43, ab11370), cardiac troponin T (CTNT, ab10214), CD68 (ab955), CD86 (ab220188), CD206 (ab64693), iNOS (ab49999), TGF-β (ab31013) and von Willebrand factor (vWF, ab6994) were ordered from Abcam. The primary antibody against alph smooth muscle actin (a-SMA, BM0002) was obtained from Boster Biological Technology (Wuhan, China). Male Sprague-Dawley (SD) rats (male, weight $220 \pm 20$ g, 7–8 weeks) were provided from the Laboratory Animal Center of the Academy of Southern Medical University (Guangzhou, China) under the ethics committee guidelines, and laboratory animals approved by Southern Medical University Animal Ethics Committee (SYXK(粤)2021-0167).

### Preparation of cellulose nanocrystals (CNC) from sea squirts

Controlled acid hydrolysis of sea squirt was used to prepare CNC[28]. Tunic of sea squirt was hydrolyzed using sulfuric acid (55 wt%). The tunic-to-acid ratio was chosen to be 1:10, (i.e., 1 g tunic:10 mL acid). Tenfold distilled water was added to the hydrolysis vessel to stop the reaction. Then, samples were centrifuged at 8000 rpm for 15 min, and the redundant solution was removed. To remove the residual sulfuric acid, hydrolyzed samples were dialyzed against deionized water for 1 week using a membrane with MWCO of 12-14KD. Finally, the sample was freeze-dried for use.

### Preparation of CNC-CHO

Firstly, 5 g cellulose was added to 200 mL 5% $NaIO_4$ solution and stirred in dark for 6 h. Secondly, 20 mL glycol was added to stop the reaction. Finally, freeze-drying was performed after repeated washing with deionized water and anhydrous ethanol.

### Fabrication of CPA hydrogel

250 mg CNC-CHO was dissolved in 5 mL NaOH/Urea solution, and was evenly dispersed by ultrasound to prepare 5% CNC-CHO solution. Afterward, the as-prepared CNC-CHO solution, 10 % PEI solution, AA and BAC (N, N'-bis(acryloyl) cystamine) were stirred to form a homogeneous solution, and then APS (ammonium persulfate) and TEMED (N,N,N',N'-Tetramethylethylenediamine) were added into the solution and homogeneously stirred to form evenly mixture. Finally, the mixture was poured into PTFE molds at 50 °C for 4 h to achieve hydrogels (denoted as CPA hydrogel).

### Fabrication of CPAM hydrogel

250 mg CNC-CHO was dissolved in 5 mL NaOH/Urea solution, and was dispersed evenly by ultrasound to prepare a 5% CNC-CHO solution. Afterward, the CNC-CHO solution, 10 % PEI solution, AA, MASEP (3-propyl methacrylate potassium) and BAC were stirred to form a homogeneous solution, and then APS and TEMED were added into the solution and homogeneously stirred to form evenly mixture. Finally, the mixture was poured into PTFE molds at 50 °C for 4 h to achieve hydrogels (denoted as CPAM hydrogel).

## Fabrication of CPAMC hydrogel

250 mg CNC-CHO was dissolved in 5 mL NaOH/Urea solution, and was dispersed evenly by ultrasound to prepare a 5% CNC-CHO solution. Afterward, the CNC-CHO solution, 10% PEI solution, AA, MASEP, CA (caffeic acid) and BAC were stirred until a homogeneous solution, and then APS and TEMED were added into the solution and homogeneously stirred to form evenly mixture. Finally, the mixture was poured into PTFE molds at 50 °C for 4 h to achieve hydrogels (denoted as CPAMC hydrogel).

## Fabrication of PAMC hydrogel

1.25% PEI solution, 15% AA, 0.2 g/mL MASEP, 1 mg/mL CA and BAC were stirred until a homogeneous solution, and then APS and TEMED were added into the solution and homogeneously stirred to form evenly mixture. Finally, the mixture was poured into PTFE molds at 50 °C for 4 h to achieve hydrogels (denoted as PAMC hydrogel).

## Fabrication of CPAMC/PCA Janus hydrogel

Firstly, CPAMC pregel was prepared according to above method. Then, 1.5 g of AA, 0.1 g CNC-COOH, 0.2 g PEGDA, and moderate APS were mixed together and was stirred to prepare PCA solution. Finally, the PCA solution was added to CPAMC pregel at 50 °C for 4 h to achieve CPAMC/PCA Janus hydrogels.

## Adhesion tests

Peeling and lap-shear adhesive strength measurements of CPAMC hydrogels were performed using universal test machine (AMETEK, America), and the values were calculated by dividing the maximum load by the corresponding overlapping area of each sample. Three samples for each group ($n = 3$) were used in adhesion test. First, the peeling tensile stress was performed with cardiac tissue substrates. The fresh porcine cardiac muscle was obtained from the market and then was cut into rectangle sections at 5.0 cm × 2.0 cm. The needless fat was shed off from the cardiac muscle pieces, and the thickness was controlled to 2–3 mm and then placed in PBS for immediate test in order to ensure the tissues were moist. CPAMC hydrogel were cut to size of 20 mm × 20 mm. CPAMC hydrogels were adhered on a piece of tissue, and the other piece tissue was pressed immediately. The samples were allowed to adhere for 10 min at 37 °C under hygrothermal condition. A tensile tester with a 50 KN load cell was used, and the samples were fixed between the two film clamps with the tensile rate was set at 5 mm/min. Then, the lap-shear tensile stress was carried out with glass substrate, similar process was performed using the standard glass slides with the size of 5.0 cm × 2.4 cm. However, the glass slide was maintained in a room temperature environment before use, and the adhesive area was 20 mm × 20 mm.

## Characterizations of different hydrogels

The morphologies of CPA, CAPM, CPAMC, and CPAMC/PCA hydrogels were observed by scanning electron microscope (SEM, H-7650, Hitachi, Japan). Fourier transform infrared spectra (FTIR) of the samples were collected with a FT-IR spectrometer (Thermo Electron Instruments Co., Ltd., USA) in the frequency range of 4000–400 cm$^{-1}$ with a total of 32 scans and resolution of 4 cm$^{-1}$. The mechanical properties of different hydrogels were detected by recycle compression test with using the universal test machine (AMETEK, America), compressive ramp up to 60% strain and strain rate of 10% per minute, preload of 0.05 N was applied.

## Electroactivity measurements

The conductivity of the hydrogels was evaluated using an electrochemical workstation (CS350H, China). Cyclic voltammetry (CV) measurements were performed in the potential range of −1.0–1.0 V at the scan rate of 100 mV/s. EIS of hydrogels were carried out at an AC amplitude of 10 mV and frequencies of 1 MHz–0.01 Hz.

## Cell viability and immunofluorescence staining of CMs on different hydrogels

Primary cultured cardiomyocytes (CMs) were isolated from the heart of 2-day-old Sprague-Dawley rats by a typical method reported by our group[1]. In short, the heart was dissected and was separated into a single-cell suspension with 0.1% type II collagenase and 0.25 % Tyrisin (Sigma). The suspension cells were then centrifuged for 5 min at 900 g and then these cells were cultured on plate for 2 h. Finally, the non-attached CMs were collected through centrifugation.

To evaluate the biocompatibility of hydrogels, live-dead staining and CCK-8 test were performed on CMs seeded on the CPA, CPAM and CPAMC hydrogel respectively. For immunofluorescence staining, the samples were incubated in primary antibodies overnight at 4 °C, including mouse anti-α-actinin (1:250), mouse CTNT (1:250) and rabbit anti-CX 43 (1:1000), and then treated with Alexa Fluor-488 Donkey anti-mouse IgG (H&L) (1:500) or Alexa Fluor-568 donkey anti-rabbit IgG (H&L) (1:500) for 1 h. The stained samples were further incubated with 4′,6-diamidino-2-phenylindole (DAPI) for 30 min and observed using confocal microscopy (Zeiss Zen software (ver. 3.2)).

## Calcium transients imaging

After being cultured for 7 days, the CMs on hydrogels were stained with Fluo-4 AM to observe the Ca$^{2+}$ activity according to the production instruction. Briefly, the samples were washed with PBS and then incubated in Fluo-4 AM work solution at 37 °C in 5 % (v/v) CO$_2$ for 45 min. Finally, the samples were washed using buffer solution, and the calcium activity were recorded under fluorescence microscope (Olympus BX53 software). The video data were analyzed by ImageJ software and Origin 8.

## Spontaneous beating analysis of the cardiomyocytes on scaffolds

The beating video of the CMs on hydrogels were recorded using a video capture program. The capture rate of video were 20 frames per second. The beating signals were analyzed using the image processing software ImageJ.

## Intracellular antioxidation experiment

The CMs were seeded on different hydrogels and cultured at 37 °C. After 500 μM H$_2$O$_2$ treated 12 h, these cells were stained by the live/dead cell staining kit. Afterwards, confocal microscopy was used to get cell images. And, after 500 μM H$_2$O$_2$ treated and cultured, the content of cellular ROS was detected and analyzed by DCFH-DA dyes stained. Meanwhile, the oxidative stress indicators (SOD, LDH and MDA) were measured, according to the corresponding schemes of test kits, respectively.

## MI model establishment, ECP implantation, and cardiac function evaluation

All animal experiments were performed under the ethics committee guidelines, and laboratory animals approved by Southern Medical University Animal Ethics Committee. SD rats (male, age 7–8 weeks, weight 220 ± 20 g) were anesthetized with isoflurane and administered with mechanical ventilation. After a left lateral thoracotomy and pericardectomy, the ligation operation of the left anterior descending artery (LAD) was performed. On day 7 after ligation, the cardiac functions of SD rats were assessed by echocardiography and the rats with fractional shortening (FS) value ≤30% were selected and taken as MI models. In the sham group ($n = 3$), only thoracotomy was performed without LAD. Then the SD rats with FS less than 30 were randomly divided into MI group ($n = 5$), CPAM ECP group ($n = 5$), CPAMC ECP ($n = 5$) group and CPAMC/PCA ECP group ($n = 4$). One week after ligation, different ECPs were adhered to the infarcted area of the MI rats in transplant groups. Meanwhile, secondary thoracotomy was conducted in the sham group and MI group just like the transplant groups.

## Echocardiography test

One week after MI model establishment, and 4 weeks after ECPs transplantation, the left ventricular (LV) functions of the rats were evaluated using the IE33 echocardiography system (Vevo2100, Visual Sonics) separately. M-mode and B-mode, which reflect LV anterior wall morphology and beating activity, were recorded using M250 transducer respectively. The correlative indexes for evaluating cardiac function, such as FS and EF, were calculated based on three consecutive cardiac cycles.

## Histological analysis of cardiac sections

Four weeks after the transplanted, all the animals were euthanized. The hearts were harvested and cut into frozen-sections with 6 μm thickness. The histologic features of MI were obtained by Masson tricolor staining following the manufacturer's instructions. The infarcted area of heart was calculated based on the ratio of collagenous area (blue) to myocardial area (red). The data were analyzed by Image J and Origin 8 software. Immunofluorescence staining procedure as following: The cardiac sections were washed 3 times in PBS, permeabilized with 0.2% Triton X-100 at room temperature for 20 min. After washed with PBS, the samples were blocked in 2% BSA at room temperature for 1 h. The primary antibodies of α-actinin (1:250), CX43 (1:1000), CD68 (1:200), CD86 (1:200), CD206 (1:200), vWF (1:250) and α-SMA (1:200) were added at 4 °C for overnight. And then the samples were incubated in secondary antibodies work solution for 2 h at room temperature. Finally, the nucleuses were marked with DAPI. The pictures were obtained using fluorescence microscope (Olympus BX53 software).

## Statistical analysis

At least three independent experiments of each type have been done and produced consistent results. Specifically, the experiments shown in the following figures were repeated three times: Figs. 2a, 4d, 5a–c, 6a, b, 7h, i, 8b–d; Supplementary Figs. 2a, 9, 10a, 12, 13, 14. All results were analyzed using SPSS20.0 and GraphPad prism 7 software. The data were expressed as means ± standard deviations (SD). Statistical analyses were performed using one-way analysis of variance (ANOVA). Tukey HSD post hoc testing was used as the post hoc correction to compare multiple groups.

## Reporting summary

Further information on research design is available in the Nature Portfolio Reporting Summary linked to this article.

## Data availability

The authors declare that all data supporting of results in this study are available within the paper and its Supplementary Information, or from the corresponding authors upon reasonable request. All data are available in the main text or the supplementary materials. Source data are provided with this paper.

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

## Acknowledgements

This research was supported by the National Natural Science Foundation of China (Grant No. U21A20173, 52003113, 31922043, 32071355, 32071363), Science and Technology Planning Project of Guangdong Province (2021A1515010745, 2020A1515110356, 2020B1212060037), Science and Technology Projects of Guangzhou City (202102020359, 201804020035), Guangzhou Science and Technology Project Key Pro-ject Topic (201904020031) and Key Research & Development Program of Bioland Laboratory (Guangzhou Regenerative Medicine and Health Guangdong Laboratory, 2018GZR110104002).

## Author contributions

H.H. and X.Q. conceived the research and designed the experiments; Y.H., Q.L., Q.D., and P.C. characterized the materials and performed the cell experiments; Y.H., Q.D., J.Z. and X.C. performed the animal experiments; Y.H., Q.L. P.C., and L.W. analyzed the data; Y.H. drafted the manuscript which was edited and revised with H.H. and X.Q.; All authors contributed to the discussion of the results at regular intervals throughout the study.

## Competing interests

The authors declare no competing interests.
