## [Peer Review File · Nature Communications]

A Smart Adhesive Janus Hydrogel for Non-Invasive cardiac repair and Tissue adhesion preventionReviewers' Comments:

Reviewer #1:

Remarks to the Author:

This manuscript is an interesting piece of work. The research of "A smart Janus adhesive hydrogel enabling on-demand and noninvasive removal as a cardiac repairing patch with postoperative tissue anti-synechia" is reported in detail. Therefore, there is sufficient significance to meet publication criteria. I suggest that this article be revised and submitted for review again.

1. It is mentioned in line 164 that "cellulose nanocrystals (CNC) ...enhanced physical interlocking for better adhesion of hydrogels...". The expression here is inappropriate. CNC cannot improve the adhesion performance. Please specify this.
2. Lines 224-226. The preceding text here is a discussion of mechanical strength and does not deal with data on elastic modulus. Yet the elastic modulus is discussed in lines 224-226, where it appears less appropriate.
3. For the discussion of mechanical properties in the article, the data of key sample CPAMC/PCA hydrogel is lacking. It is suggested to supplement the data and discuss.
4. Lines 301-308. From this statement, it can be seen that the mechanical properties and adhesion of the hydrogel increase with time, but it is essentially caused by dehydration of the hydrogel after being placed for a long time, and it can be seen that this change leads to a sharp increase in the elastic modulus of the hydrogel, which is inconsistent with the demand for the elastic modulus of natural myocardium discussed by the author in line 224 above. First of all, in the actual application scenario, the situation of long-term exposure to air does not occur. Second, the stiffness of the hydrogel caused by the sharp increase of elastic modulus does not meet the actual application requirements. Therefore, I do not understand the significance of the discussion in this paragraph for this article. The author is required to explain this.
5. Lines 349-352. For the discussion of on-demand removal in this paragraph, the author mentioned that BAC acts as a crosslinker in the hydrogel, and we know that the adhesion behavior occurs between the interface of the hydrogel and the body. However, the author said in line 349-352 that BAC fracture led to the conversion of adhesion ability. We believe that the fracture of BAC crosslinker caused the disintegration of the hydrogel structure, rather than the loss of the adhesion ability of the hydrogel. Therefore, the author needs to make a more detailed discussion here.

Reviewer #2:

Remarks to the Author:

In this manuscript, the authors developed a multifunctional Janus hydrogel conductive patch for repairing myocardial infarction and preventing inter-tissue adhesion. The idea of the multifunctional and smart adhesive design for addressing the secondary damage or tissue synechia issue is ingenious and novel. They found that the Janus cardiac patch can quickly adhere to the beating heart surface, offering the machinal support and mechanical-electronic conduction coupling for MI repair; and at the other side, it prevents postoperative tissue adhesion and acts as a physical barrier screening exogenous inflammatory factors and ROS species, which in turn is beneficial for MI repair and postoperative healing. In vivo and in vitro evidences further support the perfect therapeutic performance of this hydrogel. Given the novelty of the multifunctional Janus hydrogel and the comprehensive experimental data, I would recommend accepting this manuscript in Nature Communications with some minor issues.

1. In animal experiments, the performance of MI repair is well, however, it is fuzzy that whether or not these implantations contain cardiomyocytes. According to their previous publicized work in the same group, they usually inoculated the scaffolds with cells, which maybe confuse the real potential of this fresh developed Janus hydrogel. The authors need to indicate whether the graft was inoculated with cardiomyocytes and give a detailed discussion on these performances whatever with or without cells.

2. The performance of myocardial infarction repair in this system is nice, but compared with the previous publications, the underlying mechanism of MI repair for this hydrogel can is different or new? The authors may give a more detailed comment or discussion on these positive results.
3. Some scale bars on the image are not very clearly or missed. For example, Figure 2a, and Figure 3g.
4. More detailed description regarding the MI model should be provided. SD Rats numbers used for the in vivo experiments should be provided. Whether there were any deaths during the experiment.
5. Experimental procedure of adhesion test should be included in the methods section with more details.

REVIEWER COMMENTS

Reviewer #1 (Remarks to the Author):

This manuscript is an interesting piece of work. The research of “A smart Janus adhesive hydrogel enabling on-demand and noninvasive removal as a cardiac repairing patch with postoperative tissue anti-synechia” is reported in detail. Therefore, there is sufficient significance to meet publication criteria. I suggest that this article be revised and submitted for review again.

Response: Thank you very much for your encouraging comments and the valuable advices. We appreciate the opportunity to improve our manuscript.

1. It is mentioned in line 164 that “cellulose nanocrystals (CNC) ...enhanced physical interlocking for better adhesion of hydrogels...”. The expression here is inappropriate. CNC cannot improve the adhesion performance. Please specify this.

Response: Thanks for your professional comment. As you referred to, ordinary CNC cannot directly improve the adhesion of hydrogels, but when the CNC is modified through sulfuric acid, it can act as a connector between polymer chains and the ability of polymer chains to reorganize and dissipate energy under pressure. (Seen in Ref. 30: Rose, S. et al. Nature, 2014, 505, 16, Line 17-19 in Abstract: “Furthermore, we show that carbon nanotubes and cellulose nanocrystals that do not bond hydrogels together become adhesive when their surface chemistry is modified.”) In our work, we modified pure CNC anchoring with sulfonate and aldehyde groups by sequential treatment of sulfuric acid and NaIO₄. These sulfonated CNCs was reported to be used as functional adhesive to enhance the adhesion properties of hydrogels due to the stronger intermolecular interactions and the aldehyde group in the CNC can also form dynamic imine covalent bonds with tissue amine groups to enhance the adhesion strength between tissue and patch materials. We compared the adhesion strength of PAMC hydrogel without CNC-CHO with that of CPAMC hydrogel with CNC-CHO. It can be seen that CPAMC hydrogel can adhere to 50 g weight under water, while PAMC

hydrogel cannot stably lift 50g weight under dry environment. The lap-shear tensile results also demonstrated that the adhesion strength of CPAMC hydrogel is significantly higher than that of PAMC hydrogel. (See the line 165-169 on Page 6, line 282-290 on Page 10 and line 818-821 on the Page 30 in the revised manuscript; See Fig. S6 on Page 5 in the revised Supplementary Information).

Revisions:

...The functional cellulose nanocrystals (CNCs) anchoring with sulfonate and aldehyde groups have excellent performance in tuning mechanical properties and enhanced adhesion of hydrogels due to the for better due to their good biocompatibility, excellent mechanical properties and variability with physical interlocking, abundant intermolecular interaction sites²⁶⁻²⁸....

...In addition, to further verified the role of CNC-CHO on the adhesion of the hydrogels, we check the adhesion performance of CPAMC hydrogels with or without containing CNC-CHO. It is found that the PAMC hydrogels without CNC-CHO could not lift a weight of 50 g stably under dry environment, while the CPAMC hydrogels containing CNC-CHO could easily lift a 50 g object even under water (Fig. S6a). The lap-shear curves of the above two hydrogels also demonstrated that the adhesion strength of CPAMC hydrogel is significantly higher than that of the PAMC hydrogel (Fig. S6b), which reveals that the sulfonated and aldehyde-functionalized CNC are beneficial for the adhesive capacity of the hydrogel, which is consistent with previously reported results³⁰.....

...**Fabrication of PAMC hydrogel.** 1.25% PEI solution, 15% AA, 0.2g/mL MASEP, 1mg/mL CA and BAC were stirred until a homogeneous solution, and then APS and TEMED were added into the solution and homogeneously stirred to form evenly mixture. Finally, the mixture was poured into PTFE molds at 50 °C for 4 h to achieve hydrogels (denoted as PAMC hydrogel)

Fig. S6. a) Adhesive image of CPAMC hydrogel and PAMC hydrogel. b) Adhesion mechanical curves of CPAMC hydrogels and PAMC hydrogel to glass slide.

2. Lines 224-226. The preceding text here is a discussion of mechanical strength and does not deal with data on elastic modulus. Yet the elastic modulus is discussed in lines 224-226, where it appears less appropriate.

Response: Thanks for the insightful comments. Excellent mechanical properties, such as the suitable elasticity and good fatigue resistance, are necessary for the ideal cardiac patch scaffold. By using a stress-strain curve, the elastic modulus of the CPAMC hydrogel was determined, and the elastic modulus's differences from those of natural myocardial tissue were then examined. We have replaced the inappropriate description with elastic modulus. The changes will be highlighted in the revised manuscript. (See the line 222-229 on Page 8 and Fig. 2c on Page 7 in the revised manuscript)

Revisions:

...The elastic modulus and mechanical strength of hydrogel can be significantly improved by the interpenetrating network formed by MASEP, which elastic modulus increased from 156.86 ± 29.7 kPa to 224.4 ± 3.13 kPa in a 60 % compress strain after adding MASEP (Fig. 2c). However, the elastic modulus of CPAMC hydrogel reduced to 45 ± 1.5 KPa after introducing caffeic acid. The elastic modulus of all hydrogels is within the elastic modulus range of native myocardium³⁵, meeting the requirements of elasticity and fatigue resistance of scaffolds for cardiac tissue engineering....

Fig. 2 (c) The elasticity modulus of CPA hydrogel, CPAM hydrogel and CPAMC hydrogel (n=3)

3. For the discussion of mechanical properties in the article, the data of key sample CPAMC/PCA hydrogel is lacking. It is suggested to supplement the data and discuss.

Response: Thanks for your professional comment. Considering that the thin layer of PCA hydrogel has little influence on the overall mechanical properties of CPAMC/PCA hydrogel, and the mechanical properties of CPAMC/PCA hydrogel are similar to those of CPAMC hydrogel, the mechanical properties of CPAMC/PCA hydrogel were not tested again in original manuscript. To better answer your questions, we have added mechanical tests of CPAMC/PCA hydrogel. As expected, the mechanical properties of CPAMC/PCA hydrogel (49.53 ± 2.48 KPa) were similar to those of CPAMC hydrogel (45 ± 1.5 KPa), and their fatigue resistance was not much different, which indicated that the addition of PCA hydrogel layer would not affect the overall mechanical properties of the hydrogel, and the CPAMC/PCA hydrogel was consistent with the mechanical properties of the cardiac patch scaffold requirements (Fig. S6). And we supplemented the new comments and data of mechanical properties of CPAMC/PCA hydrogel. (See the line 395-401 on Page 14 in the revised manuscript; See Fig. S7 on Page 5 in the revised Supplementary Information)

Revisions:

...We also checked the overall mechanical properties of CPAMC/PCA Janus hydrogels, and found that the elastic modulus and fatigue resistance of CPAMC/PCA hydrogels (49.53 ± 2.48 KPa) exhibit no obvious difference from that of CPAMC hydrogels (45 ± 1.5 KPa) (Fig. S7). These results indicate that the thin PCA anti-adhesion hydrogel layer had almost no impact on the mechanics of CPAMC hydrogel, and the elastic modulus of CPAMC/PCA hydrogel is still within the range of natural myocardium, which meets the mechanical requirements of cardiac tissue engineering for elasticity....

Fig. S7. a) The stress-strain curves of CPAMC hydrogel and CPAMC/PCA hydrogel. (b) The elastic modulus of CPAMC hydrogel and CPAMC/PCA hydrogel (n=3).

4. Lines 301-308. From this statement, it can be seen that the mechanical properties and adhesion of the hydrogel increases with time, but it is essentially caused by dehydration of the hydrogel after being placed for a long time, and it can be seen that this change leads to a sharp increase in the elastic modulus of the hydrogel, which is inconsistent with the demand for the elastic modulus of natural myocardium discussed by the author in line 224 above. First of all, in the actual application scenario, the situation of long-term exposure to air does not occur. Second, the stiffness of the hydrogel caused by the sharp increase of elastic modulus does not meet the actual application requirements. Therefore, I do not understand the significance of the discussion in this paragraph for this article. The author is required to explain this.

Response: We thank the reviewer for the insightful comments. We agree that your opinion that the enhancement behavior of adhesion and mechanical properties is caused by the dehydration of the hydrogel, in our manuscript we also mentioned this point, however, we believe the introduction of dynamic covalent crosslinking system is also responsible for this biomimetic adhesion-stiffening behavior. In this work, we speculate that the adhesion-stiffening behavior of CPAMC hydrogels may be due to topographic mechanics and structural adaptations induced by the dynamically cross-linked hydrogel network, which mimics the biological strain-stiffening behavior based on the fiber network mechanics confined by the tightly packed cells of biological tissues (Ref. 36-39: Nature 2017; Nature 2019; Science 2018; Lab. Chip. 2017). When adhered to wet tissues, the formation of the chemically covalent linkages and noncovalent physically continues to gradually evolve in the CPAMC hydrogel. Along with the progressively adjustment and enhancement of interpenetrating polymeric network, the embedded water was expelled and eventually, the hydrogel remains firmly attached to the tissue (Fig. 8).

The hydrogel not only securely attached to the heart following the in-vivo implantation of ECPs after 4 weeks, but it also did not dramatically harden, and its elasticity was not significantly different from that before the transplant. This may be ascribed to the reassembly of BAC responsive crosslinking network. These results indicate that the rearrangement of hydrogel molecular chain caused by BAC dynamic cross-linking system can make the hydrogel adaptively adjust adhesion under physiological conditions. Although the mechanical properties have been enhanced, they can still meet the elastic requirements of myocardial tissue. We have added the new comments in the revised manuscript. (See the line 341-348 on Page 12 and line357-359 on Page 13, line 709-713 on the Page 26 in the revised manuscript)

To further confirm your question, we added a control experiment of non-responsive covalent crosslinking system with MBA as crosslinker, as shown in the following Figure R1. The adhesion strength of PAMC hydrogel with MBA crosslinker increased from 187.7 KPa (day 1) to 1127.18 KPa (day 7), indicating that the enhancement of adhesion may be caused by dehydration of hydrogel. However, compared with that, the

adhesion strength of CPAMC hydrogel has increased more significantly, from 223.14 KPa to 1231.44 KPa, which indicates that in addition to the adhesion enhancement caused by dehydration of hydrogel, the role of BAC dynamic crosslinking system cannot be ignored.

Figure R1 (a) Adhesion properties of CPAMC hydrogel with BAC cross-linking and CPAMC hydrogel with MBA cross-linking on day1. (b) Adhesion properties of CPAMC hydrogel with BAC cross-linking and CPAMC hydrogel with MBA cross-linking on day7. (c) Statistical analysis of adhesive strength of CPAMC hydrogel with BAC cross-linking and CPAMC hydrogel with MBA cross-linking in glass in air condition with different contact time.

Revisions:

...In the current system, at one hand, the dynamic rearrangement induced by the physiologically responsive disulfide bond in the hydrogel could gradually extrude the residue water, which leads to the adhesion and mechanical property enhancement behavior after the hydrogel being adhered for a relatively long time. At the other hand, from our in-vivo implantation result, the firmly adhesion of wet tissue after 4 weeks indicates that this dynamical reconstruction of interpenetrating network and adaptive mechanical stiffening within the responsive hydrogel to the MI microenvironment is also responsive for the enhanced adhesion capacity.....

..., in particular with the smart and adaptive adhesion requirements for long-term repair of heart and other tissues accompanied by dynamically motion or mechanical movement.....

...This is attributed to the fact that the molecular chain rearrangement of the adhesive layer hydrogel caused by the BAC dynamic cross-linking system can enable the

hydrogel to adaptively regulate adhesion under physiological conditions, while the PCA anti-adhesive layer reduces the aggregation of inflammatory cells and adhesion molecules.....

5. Lines 349-352. For the discussion of on-demand removal in this paragraph, the author mentioned that BAC acts as a crosslinker in the hydrogel, and we know that the adhesion behavior occurs between the interface of the hydrogel and the body. However, the author said in line 349-352 that BAC fracture led to the conversion of adhesion ability. We believe that the fracture of BAC crosslinker caused the disintegration of the hydrogel structure, rather than the loss of the adhesion ability of the hydrogel. Therefore, the author needs to make a more detailed discussion here.

Response: We thank the referee for these professional comments. In the construction of CPAMC hydrogel, we designed multiple cross-linking systems. In addition to the chemical cross-linking formed by BAC, there are also dynamic cross-linking between aldehyde group and amino group and hydrogen bond interaction between cellulose and acrylic acid carboxyl group. These chemical and physical cross-linking together form stable CPAMC hydrogel. Furthermore, the disulfide bond in BAC, as a dynamic bond, breaks in the oxidative environment, but when returning to the physiological state, the disulfide bond will form disulfide bond again, which will rearrange the hydrogel to adapt to the new structure. Therefore, we can see that the morphology of CPAMC hydrogel after GSH treatment does not change significantly (Fig. 3g and Movie S4).

Reviewer #2 (Remarks to the Author):

In this manuscript, the authors developed a multifunctional Janus hydrogel conductive patch for repairing myocardial infarction and preventing inter-tissue adhesion. The idea of the multifunctional and smart adhesive design for addressing the secondary damage or tissue synechia issue is ingenious and novel. They found that the Janus cardiac patch can quickly adhere to the beating heart surface, offering the machinal support and mechanical-electronic conduction coupling for MI repair; and at the other side, it prevents postoperative tissue adhesion and acts as a physical barrier screening

exogenous inflammatory factors and ROS species, which in turn is beneficial for MI repair and postoperative healing. In vivo and in vitro evidences further support the perfect therapeutic performance of this hydrogel. Given the novelty of the multifunctional Janus hydrogel and the comprehensive experimental data, I would recommend accepting this manuscript in Nature Communications with some minor issues.

Response: Thank you for improving our manuscript by your comments. We are grateful to get this positive and encouraging feedback.

1. In animal experiments, the performance of MI repair is well, however, it is fuzzy that whether or not these implantations contain cardiomyocytes. According to their previous publicized work in the same group, they usually inoculated the scaffolds with cells, which maybe confuse the real potential of this fresh developed Janus hydrogel. The authors need to indicate whether the graft was inoculated with cardiomyocytes and give a detailed discussion on these performances whatever with or without cells.

Response: Thanks for your professional comments. Just as you commented, we did not inoculate with cardiomyocytes in this work for better understanding the potential of this fresh developed Janus hydrogel.

In process of MI repair, the scaffolds are mainly aimed at providing mechanical support to the infarcted area, which minimizes cardiac remodeling and helps preserve the contractile function of the heart. And implanted cells can produce and release various cardiac regenerative and immunomodulatory cytokines to promote MI repair through paracrine effect in previous studies (Ref.1: Wang, L. et al. Nat. Biomed. Eng. 2021, 5; Ref.4: Song, X. et al. Biomaterials, 2021, 273; Ref.8: He, Y. et al. Bioact. Mater. 2021, 6). However, pure scaffolds without implanted cells also can accelerate the restoration of myocardial infarction by providing good mechanics and reconstruction of electric conduction coupling between infarction area and normal cardiac tissue (Ref 12: Liang, S. et al. Adv. Mater. 2018, 30, e1704235; Ref 14: Zhou, J. et al. Adv. Sci. 2021, 8, e2100505; Walker, et al., Biomaterials, 2019, 207:89).

In this work, we focus on studying the effect of pure Janus hydrogel on the MI repair performance, and our Janus hydrogel not only had good electrical conductivity, mechanical properties, anti-inflammatory and antioxidant properties, but also had excellent wet surface adhesion and anti-adhesion. The excellent properties of electrical conductivity, mechanical properties and antioxidant properties of Janus hydrogel could improve the physicochemical microenvironment after MI, reduce the oxidative damage and inflammatory damage of cardiomyocytes, promote the regeneration of a large number of blood vessels, and reduce the fibrosis in the infarcted area of rats even. Furthermore, the outstanding wet tissue adhesion and anti-adhesion of Janus hydrogel reduced the second tissue damage and inflammatory invasion and tissue adhesion, which effectively promotes the recovery of cardiac function. And these results were verified by a series of *in-vitro* and *in-vivo* experiments shown in Fig. 6, Fig. 7 and Fig. 8, demonstrating their perfect MI repair performance as engineering cardiac patches. We have added the new comments in the revised manuscript. (See the line 565-569 on Page 21, and line 739-744 on Page 27 in the revised manuscript)

Revisions:

...Considering the physicochemical microenvironment of Janus hydrogel *in vitro* can promote the maturation and functionalization of CMs, and can reduce the oxidative damage of CMs and promote the polarization of reparative inflammatory cells, we further studied *in vivo* whether this microenvironment can promote the repair of myocardial infarction injury without the participation of exogenous cells.....

...These results indicate that without the paracrine effect of exogenous cells, Janus hydrogel can provide excellent physicochemical microenvironment, reconstruct the electrical conduction in the myocardial infarction area, protect CMs from oxidative damage and inflammatory damage, promote the regeneration of blood vessels in the myocardial infarction area, and avoid the secondary damage and tissue adhesion caused by traditional patch transplantation, so as to effectively promote the repair of damage after myocardial infarction.....

2. The performance of myocardial infarction repair in this system is nice, but compared with the previous publications, the underlying mechanism of MI repair for this hydrogel can be different or new? The authors may give a more detailed comment or discussion on these positive results.

Response: Thanks for your insightful comments. We think the underlying mechanism of MI repair for this hydrogel can be different. In addition to the repair function of traditional cardiac patch, the excellent tissue adhesion of Janus hydrogel avoids the secondary tissue damage during patch fixation and the subsequent tissue adhesion.

Firstly, a large number of literatures have proved that elastic conductive scaffolds can contribute to MI repair by providing mechanical support and reconstructing electrical conduction between myocardial infarction area and non-myocardial area (Malki, et al. *Nano Lett.* 2018,18: 4069; Walker, et al. *Biomaterials*, 2019, 207:89; Jiang, et al. *Adv. Mater.* 2022, 34: 220141). Therefore, our Janus hydrogel with good elasticity and conductivity can provide mechano-electrical coupling microenvironment for infarct area tissue to accelerate MI repair. Secondly, ROS generated after myocardial infarction will continuously damage cardiomyocytes, resulting in the continuous expansion of myocardial infarction area (Choe, et al., *Biomaterials*, 2019, 225:119513; Shilo, et al., *Adv. Sci.* 2021,8:2102919). Hence, we can reduce the CMs damage, inhibit the expansion of infarction area, and facilitate the MI repair by removing ROS in the infarction area. In addition, we can reduce the inflammatory damage by promoting the polarization of M2-type macrophages, and accelerate the massive generation of blood vessels to restore blood supply, thus promoting the MI repair. More importantly, the adhesiveness of Janus hydrogel avoids the secondary injury of myocardial tissue caused by traditional suture fixation, and the existence of the anti-adhesive layer blocks the inflammatory damage and tissue adhesion caused by the surrounding tissues of the heart. In conclusion, the excellent performance of Janus hydrogel can accelerate tissue repair after myocardial infarction. The new comments are now included in the revised manuscript. (See the line 589-592 on Page 22 and line 644-647 Page 24 in the revised manuscript)

Revisions:

...Additionally, the excellent tissue adhesion of hydrogel scaffold also is beneficial for the minimization of secondary harm during transplantation and further safeguards of the injured cardiac function during and after healing....

...More importantly, the presentation of anti-sticky layer hinders the invasion of foreign inflammatory cells and inflammatory factors, and the remarkable tissue adherence of the hydrogel scaffold significantly reduces secondary injury after transplantation as well as inflammation brought on by injury as well....

3. Some scale bars on the image are not very clearly or missed. For example, Figure 2a, and Figure 3g.

Response: Thanks for your nice suggestion. We have provided the clearer scale bars in the Fig. 2a and added new scale bars in Fig. 3g accordingly. (See the Fig. 2a on Page 7 and Fig. 3g on Page 11 in the revised manuscript)

4. More detailed description regarding the MI model should be provided. SD Rats numbers used for the in vivo experiments should be provided. Whether there were any deaths during the experiment.

Response: Thanks for your professional and careful comments. We have supplemented the description of establishing the MI model in the **Materials and methods** portion of the revised manuscript (Page 34 in the revised manuscript, highlighted in red). In the whole animal experiment, no rats died from the postoperation to the end of the repair, except for the rats that died during the thoracotomy.

The added statement including the number of SD rats used for this experiment is as below: "SD rats (male, age 7-8 weeks, weight 220±20g) were anesthetized with isoflurane and administered with mechanical ventilation. After a left lateral thoracotomy and pericardectomy, the ligation operation of the left anterior descending artery (LAD) was performed. On day 7 after ligation, the cardiac functions of SD rats were assessed by echocardiography and the rats with fractional shortening (FS) value ≤ 30% were selected and taken as MI models. In the sham group (n=3), only thoracotomy was performed without LAD. Then the SD rats with FS (short axial

shortening) less than 30 were randomly divided into MI group (n=4), CPAM ECP group (n=5), CPAMC ECP (n=5) group and CPAMC/PCA ECP group (n=4).”

5. Experimental procedure of adhesion test should be included in the methods section with more details.

Response: Thanks for your professional suggestion. According to your suggestion, the corresponding details and information was added in the paragraph of “Adhesion Tests.” in **Materials and methods** portion of the revised manuscript. (See the last paragraph on Page 28 and the first paragraph on Page 29 in the revised manuscript). The added experimental procedure is “Peeling and lap-shear adhesive strength measurements of CPAMC hydrogels were performed using universal test machine (AMETEK, America) and the values were calculated by dividing the maximum load by the corresponding overlapping area of each sample. Three samples for each group (n=3) were used in adhesion test. First, the peeling tensile stress was performed with cardiac tissue substrates. The fresh porcine cardiac muscle was obtained from the market and then was cut into rectangle sections at 5.0 cm × 2.0 cm. The needless fat was shed off from the cardiac muscle pieces, and the thickness was controlled to 2~3 mm and then placed in PBS for immediate test in order to ensure the tissues were moist. CPAMC hydrogel were cut to size of 20 mm × 20 mm. CPAMC hydrogels were adhered on a piece of tissue and the other piece tissue was pressed immediately. The samples were allowed to adhere for 10 min at 37 °C under hygrothermal condition. A tensile tester with a 50 KN load cell was used and the samples were fixed between the two film clamps with the tensile rate was set at 5 mm/min. Then, the lap-shear tensile stress was carried out with glass substrate, similar process was performed using the standard glass slides with the size of 5.0 cm × 2.4 cm. However, the glass slide was maintained in a room temperature environment before use and the adhesive area was 20 mm × 20 mm.”

Reviewers' Comments:

Reviewer #1:

None

Reviewer #2:

Remarks to the Author:

I think that this revised manuscript could be accepted for publication now.

REVIEWERS' COMMENTS

Reviewer #2 (Remarks to the Author):

I think that this revised manuscript could be accepted for publication now.

Response: We thank the reviewer for the comments.